# Generation and maintenance of the circularized multimeric IS26-associated translocatable unit encoding multidrug resistance

Masamune Aihara [1,2,8] ✉, Yasuhiro Gotoh [3,8], Saki Shirahama[1], Yuichi Matsushima [4,5], Takeshi Uchiumi [2], Dongchon Kang[5,6,7] & Tetsuya Hayashi [3]

In gram-negative bacteria, IS26 often exists in multidrug resistance (MDR) regions, forming a pseudocompound transposon (PCTn) that can be tandemly amplified. It also generates a circular intermediate called the "translocatable unit (TU)", but the TU has been detected only by PCR. Here, we demonstrate that in a *Klebsiella pneumoniae* MDR clone, mono- and multimeric forms of the TU were generated from the PCTn in a preexisting MDR plasmid where the inserted form of the TU was also tandemly amplified. The two modes of amplification were reproduced by culturing the original clone under antimicrobial selection pressure, and the amplified state was maintained in the absence of antibiotics. Mono- and multimeric forms of the circularized TU were generated in a RecA-dependent manner from the tandemly amplified TU, which can be generated in RecA-dependent and independent manners. These findings provide novel insights into the dynamic processes of genome amplification in bacteria.

Insertion sequences (ISs) are small transposable elements that encode only transposase (TPase) genes. They play important roles in the diversification and evolution of prokaryotes by gene inactivation, inducing various genomic rearrangements, and directing gene sequestration as a prologue to horizontal gene transfer[1]. More than 4600 types of ISs have been identified, which are classified into 29 families based on their sequence similarities and the chemistry used by their TPases[2], as listed in the ISfinder database[3] and the TnCentral database[4] and described in TnPedia (https://tncentral.ncc.unesp.br/TnPedia/).

Among the large variety of ISs, the clinical importance of IS26 and its relatives belonging to the IS6 family has been emphasized due to their frequent participation in the dissemination of antimicrobial resistance (AMR) genes by gathering them in gram-negative bacterial genomes, leading to the emergence of multidrug-resistant (MDR) strains[5]. Multiple copies of IS26 are often present in MDR regions, and directly oriented IS26 copies form a compound transposon called a pseudocompound transposon (PCTn) because IS6 family members transpose from a donor to a recipient DNA by forming a cointegrate[6,7]. In addition, the tandem amplification of a part of PCTn, comprising one copy of IS26 and the passenger sequence captured by PCTn, was observed in several clinical isolates, enhancing their MDR phenotypes and resulting in therapy failure, recurrent bacteraemia, and nosocomial outbreaks[8–12]. As a mechanism underlying this amplification, the formation of a circular intermediate comprising a single IS26 copy and the passenger sequence, which was named the 'translocatable unit (TU)', has been proposed[10]. In this model, a circular form of TU is generated from the PCTn and inserted into a preexisting IS26 copy[13–17]. If reinsertion occurs at one end of the PCTn, the TU is amplified in tandem there. However, nonreciprocal recombination (unequal crossover) at the IS26 copies can also lead to the tandem amplification of a segment corresponding to the TU[18]. It should also be noted that the presence of a circular TU

¹Department of Clinical Chemistry and Laboratory Medicine, Kyushu University Hospital, Fukuoka, Japan. ²Department of Health Science, Graduate School of Medical Sciences, Kyushu University, Fukuoka, Japan. ³Department of Bacteriology, Graduate School of Medical Sciences, Kyushu University, Fukuoka, Japan. ⁴Department of Biological Sciences, Graduate School of Science, Osaka University, Toyonaka, Japan. ⁵Department of Clinical Chemistry and Laboratory Medicine, Graduate School of Medical Sciences, Kyushu University, Fukuoka, Japan. ⁶Kashiigaoka Rehabilitation Hospital, Fukuoka, Japan. ⁷Department of Medical Laboratory Science, Faculty of Health Sciences, Junshin Gakuen University, Fukuoka, Japan. ⁸These authors contributed equally: Masamune Aihara, Yasuhiro Gotoh. ✉e-mail: aihara.masamune.402@m.kyushu-u.ac.jp

molecule has been detected only by PCR for Tn*4532*B, a derivative of Tn*4532*, containing two additional bases inserted at the right end of the upstream IS*26* copy[14].

We recently identified five *Klebsiella pneumoniae* isolates (KpWEA1, KpWEA2, KpWEA3, KpWEA4-1, and KpWEA4-2) that were consecutively isolated from a patient who received heavy antimicrobial treatment[19] (Fig. 1A). The isolates showed gradual enhancement of the MDR phenotype (Fig. 1A). By analysing their complete genome sequences, we showed their clonality and stepwise accumulation of chromosomal mutations that lowered drug permeabilities. In addition, these isolates showed intriguing changes in extrachromosomal elements during within-host evolution (Fig. 1A). While the first isolate, KpWEA1, already possessed a 220-kb resistance plasmid, P1, KpWEA3 additionally acquired the P2 and P3 elements. Although P3 was found to be a so-called plasmid prophage and transiently present in this clone, P2 remained persistent in the latter isolates, KpWEA4-1 and KpWEA4-2. Notably, the P2 sequence (36.5 kb) was identical to a part of an IS*26*-associated PCTn in P1 (P2 in plasmid P1,

referred to as pP2 in this manuscript). It comprised a complete copy of IS*26*, which was directly followed by a partial IS*26* copy and a 35.5-kb sequence that was carried by the PCTn. While the 35.5-kb sequence encoded various genes, including several AMR genes (Fig. 1B), no known replicons were found. Thus, it was unknown how the circular P2 element (referred to as cP2) was inherited by the descendants of KpWEA3 (KpWEA4-1 and KpWEA4-2).

After publishing these results, we realized that cP2 represents a TU-like structure comprising a complete copy of IS*26*, which was directly followed by a partial IS*26* copy (193 bp) and a 35.5-kb passenger sequence (referred to as PS in this manuscript) of the PCTn. Because the TU could be amplified in tandem as described above, we hypothesized that the tandem amplification of pP2 occurred in these isolates, which might be involved in the generation of a stably inheritable cP2 molecule. In this study, to verify this hypothesis, we reanalysed the *K. pneumoniae* isolates. By analysing the copy numbers of P2, long-read sequencing and pulsed-field gel electrophoresis (PFGE)-based profiling of extrachromosomal elements, we revealed the tandem

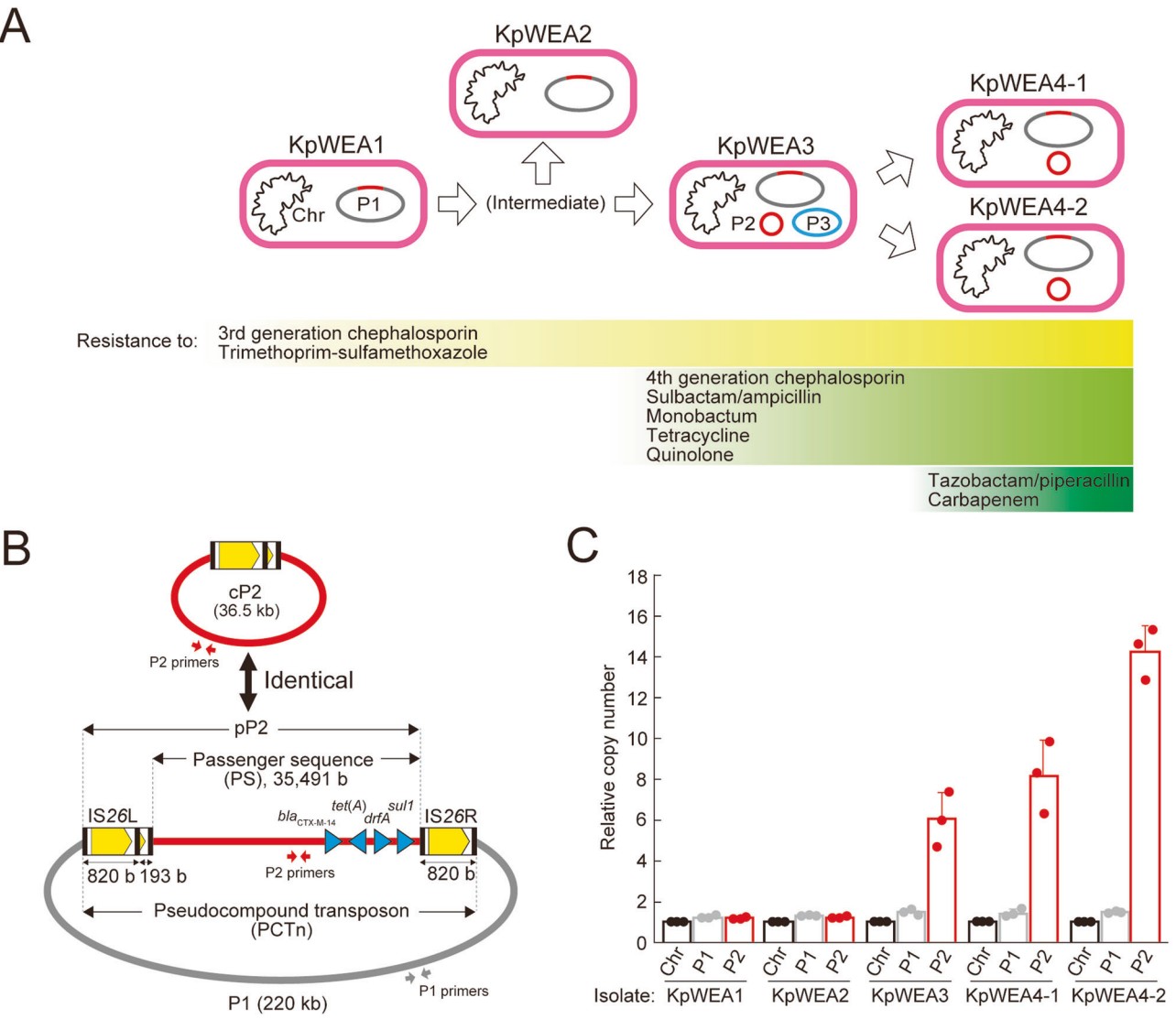

**Fig. 1 | Change in extrachromosomal contents during the within-host evolution of a *K. pneumoniae* clone. A** KpWEA1, KpWEA2, KpWEA3, KpWEA4-1 and KpWEA4-2 were isolated from a single patient in this order. Sequential acquisition of resistance to the indicated antimicrobials by these isolates was also shown. **B** Schematic presentation of the relationship of the P1 plasmid, pP2, cP2, and the IS*26*-associated pseudocompound transposon (PCTn). The passenger sequence (PS) of PCTn is indicated by red lines, and the four resistance genes in PS are

indicated by blue arrowheads. The TPase-coding gene and terminal inverted repeats of IS*26* are indicated in yellow and black, respectively. The positions of PCR primers for the specific amplification of P1 and P2 (cP2 and pP2) are indicated by the grey and red arrows, respectively. **C** The copy numbers of P1 and P2 sequences relative to that of the chromosome (Chr) in the five *K. pneumoniae* isolates. The values represent the mean ± standard deviation (s.d.) from three independent experiments.

amplification of pP2, the presence of mono- and multimeric forms of cP2, and the stable maintenance of these molecules even under antimicrobial stress-free conditions. We further demonstrate that the tandem amplification of pP2 and the generation of mono- and multimeric forms of cP2 in *K. pneumoniae* can be reproduced in vitro under selective antimicrobial pressure and provide evidence showing that mono- and multimeric cP2 molecules are generated in a RecA-dependent manner. Moreover, by constructing plasmids carrying the mini-PCTn or its TPase-deficient derivative, we show that the tandem amplification of mini-pP2 can occur in the RecA/TPase-independent manner, albeit at a low efficiency. Based on the results obtained, a possible mechanism for the generation and maintenance of the amplified state of P2 is also proposed.

## Results

### Structure of the pP2-containing region in plasmid P1

In plasmid P1, the pP2 sequence is directly followed by a complete copy of IS*26*. Thus, this region represents a PCTn, in which a partial sequence of IS*26* corresponding to the 3'-terminal sequence of IS*26* and a 35,491-bp PS region are flanked by two directly oriented IS*26* copies (Fig. 1B). These two IS*26* copies (referred to as IS*26*L and IS*26*R in the PCTn configuration) were identical in sequence. The genes encoded in PS included four AMR genes: *bla*CTX-M-14 encoding an extended-spectrum β-lactamase, *tet(A)*, *drfA*, and *sul1*. P1 additionally contained 4 complete IS*26* copies, which potentially formed additional PCTn structures (Supplementary Fig. 1). However, no direct target DNA repeats were detected for any of the complete IS*26* copies and potential PCTns, including the P2-related one. This suggests that some genome rearrangements have occurred in the regions surrounding these elements after their insertion. Notably, no IS*26* copies were detected in the chromosomes of all the *K. pneumoniae* isolates except for KpWEA4-2, in which an IS*Ecp1*-mediated translocation of a 29.5-kb segment containing IS*26*L to the chromosome was observed[19].

### Amplification of P2-related sequences

To determine the copy numbers of P2-related sequences in the five *K. pneumoniae* isolates, single colonies of each isolate were cultured in antibiotic-free lysogeny broth (LB) overnight. Then, their genomic DNA was extracted, and the copy numbers of P2-related and P1 sequences relative to the chromosome sequence were determined by digital PCR using primers and probes specific to each sequence (Fig. 1B; see Supplementary Table 1 for the primer and probe sequences). The relative copy number of P1 was consistent among the five isolates and nearly the same as that of the chromosome (Fig. 1C). In contrast, the copy number of the P2-related sequence (derived from pP2 and/or cP2) varied among the isolates. While the relative copy number of the P2-related sequence was the same as that of P1 in KpWEA1 and KpWEA2, it notably increased in KpWEA3, KpWEA4-1 and KpWEA4-2 (6.0-, 8.1-, and 14.2-fold on average, respectively) (Fig. 1C). This result indicates that the P2-related sequence was amplified in these three isolates and that the amplified state was maintained in the absence of antimicrobial pressure, at least during overnight culture.

### Detection of the tandem amplification of pP2 by long-read sequencing

Amplification of the P2-related sequence could be achieved by tandem amplification of pP2. We therefore reanalysed the long reads previously obtained from five isolates by Oxford Nanopore Technology (ONT) sequencers to identify the reads encompassing the junctions of tandemly repeated pP2 sequences and/or the pP2/chromosome junctions. Mapping of ONT reads ( > 10 kb) to the 35.5-kb PS revealed the tandem amplification of pP2 in KpWEA3, KpWEA4-1, and KpWEA4-2, as many ONT reads from these isolates contained the sequences upstream of IS*26*L or downstream of IS*26*R and tandemly repeated pP2 sequences ("Reads from pP2 derivatives" in Fig. 2). From KpWEA4-1, we obtained six ONT reads derived from a sequence where pP2 was repeated at least three times (Fig. 2). In this analysis, many reads that contained multiple copies of only the P2 sequence arranged in tandem were also obtained from the three isolates.

However, these types of reads could have been derived from not only tandemly repeated pP2 but also multimers of cP2 if such forms of cP2 were present ("Reads from pP2 or cP2 derivatives" in Fig. 2).

### Detection of tandemly repeated pP2 and multimeric forms of cP2

To further verify the tandem amplification of pP2 in plasmid P1 and examine the presence of multimeric forms of cP2, we profiled the extrachromosomal elements of each isolate by PFGE. Single colonies of each isolate were cultured in antibiotic-free LB, and extrachromosomal DNA was obtained from the cells by alkaline-SDS extraction. Note that the concentration of DNA extracted from KpWEA4-2 was always several times higher than those from the other isolates, probably due to the difference in extraction efficiency among the isolates. In the PFGE analysis of extracted DNA (Fig. 3A), that of KpWEA1 and KpWEA2 showed a single high-molecular-weight (HMW) band. As this band was also present in the other three isolates, it likely corresponds to P1 (220 kb), although the intensity of the band in KpWEA4-1 was weak. In KpWEA3, KpWEA4-1, and KpWEA4-2, multiple additional bands were observed. Among these, one band detected only in KpWEA3 was most likely the P3 prophage plasmid (110.8 kb). In the three isolates, two or three faint bands were commonly observed at positions lower than the putative P3 band. In KpWEA4-2, one clear band and one faint band were also observed at positions higher than the putative P1 band. Although these bands also appeared to be present in KpWEA3 and KpWEA4-1, their presence could not be clearly confirmed in this analysis due to the low intensities of these bands. As this difference was likely due to the difference in the aforementioned extraction efficiency of HMW DNA among the isolates, we decided to use KpWEA4-2 in the subsequent experiments to analyse the extrachromosomal contents.

To identify the origins of each band detected in KpWEA4-2 (indicated by three arrows and three open arrowheads in Fig. 3A), the extrachromosomal DNA was digested by the restriction endonucleases SpeI and XbaI and/or Exonuclease V (ExoV). SpeI/XbaI double digestion of P1 was expected to produce four long fragments (76.5, 54.2, 39.6, and 30.3 kb) and seven short fragments ( < 5.5 kb), among which the longest fragment included the entire pP2 region. ExoV treatment removes linear DNA molecules. As shown in the left panel in Fig. 3B, the bands detected in the untreated sample (Fig. 3A) were still detected after treatment with ExoV only, indicating that they were circular DNA molecules. By SpeI/XbaI digestion, the HMW bands, including a putative P1 band (arrows in Fig. 3B), disappeared, and four bands appeared at the 76.5-, 54.2-, 39.6-, and 30.3-kb positions. However, the intensity of the 76.5-kb band (a closed arrowhead at the lowest position in Fig. 3B) was much weaker than that of the other three bands. Instead, two additional bands larger than the 76.5-kb band (at approximately 110 and 150 kb; also indicated by closed arrowheads in Fig. 3B) were generated by SpeI/XbaI digestion, and the intensity of the 110-kb band was stronger than that of the 76.5-kb band. All these bands generated by SpeI/XbaI digestion disappeared after ExoV treatment, indicating that they were linear DNA molecules probably derived from P1. Moreover, among the three SpeI/XbaI digestion products indicated by closed arrowheads in Fig. 3B, the estimated size of the smallest one corresponded to the size of the P1 fragment containing one copy of pP2 (76.5 kb), and those of the two larger ones roughly corresponded to the sizes of P1 fragments that contained two or three copies of pP2 in tandem (113 and 149.5 kb, respectively). In contrast, the three faint bands indicated by open arrowheads in Fig. 3B remained after SpeI/XbaI and ExoV digestion. Thus, they were circular DNA molecules, most likely corresponding to a mono-, di-, or trimeric form of cP2.

To verify these hypotheses, we performed a Southern blotting and hybridization analysis of the same set of samples using a probe specific to the *bla*CTX-M-14 gene located in the P2 region (the right panel in Fig. 3B). This analysis revealed that all bands that we hypothesized as being derived from pP2- or cP2-related molecules yielded hybridization signals. Moreover, above the signal of the putative trimeric cP2, we detected an additional signal that was likely derived from tetrameric cP2. Finally, we performed additional digestion of the SpeI/XbaI/ExoV-treated sample using the restriction

| Configuration pattern | Copy number of PS | KpWEA1 1,489 / 511,384 | KpWEA2 1,711 / 492,635 | KpWEA3 1,370 / 86,071 | KpWEA4-1 1,489 / 45,424 | KpWEA4-2 1,713 / 76,561 |
|---|---|---|---|---|---|---|
| **(1) Reads from pP2 derivatives** | | | | | | |
|  | 1 | 21 | 22 | 0 | 2 | 1 |
|  | ≥1 | 513 | 620 | 189 | 191 | 204 |
|  | ≥1 | 0 | 3 | 0 | 0 | 1 |
|  | ≥2 | 0 | 0 | 3 | 24 | 10 |
|  | ≥2 | 0 | 0 | 0 | 1 | 0 |
|  | ≥3 | 0 | 0 | 0 | 1 | 0 |
|  | ≥1 | 517 | 592 | 168 | 160 | 196 |
|  | ≥1 | 0 | 1 | 1 | 0 | 1 |
|  | ≥2 | 0 | 0 | 7 | 18 | 15 |
|  | ≥3 | 0 | 0 | 0 | 6 | 0 |
| **(2) Reads from pP2 or cP2 derivatives** | | | | | | |
|  | ≥1 | 399 | 415 | 445 | 550 | 802 |
|  | ≥1 | 15 | 25 | 30 | 36 | 29 |
|  | ≥1 | 0 | 0 | 3 | 7 | 6 |
|  | ≥1 | 22 | 28 | 34 | 33 | 47 |
|  | ≥1 | 2 | 5 | 8 | 5 | 3 |
|  | ≥1 | 0 | 0 | 0 | 1 | 0 |
|  | ≥1 | 0 | 0 | 414 | 704 | 255 |
|  | ≥2 | 0 | 0 | 52 | 101 | 39 |
|  | ≥2 | 0 | 0 | 3 | 3 | 2 |
|  | ≥2 | 0 | 0 | 1 | 2 | 2 |
|  | ≥2 | 0 | 0 | 1 | 12 | 6 |
|  | ≥2 | 0 | 0 | 0 | 0 | 2 |
|  | ≥2 | 0 | 0 | 11 | 83 | 73 |
|  | ≥3 | 0 | 0 | 0 | 19 | 12 |
|  | ≥3 | 0 | 0 | 0 | 0 | 3 |
|  | ≥3 | 0 | 0 | 0 | 8 | 3 |
|  | ≥4 | 0 | 0 | 0 | 2 | 1 |

— Upstream region of IS26L  — Downstream region of IS26R  ☐ IS26  ▯ 3′-terminal seqeunce (193 bp) of IS26  — Passenger sequence (PS)

**Fig. 2 | The numbers of ≥10 kb P2-related ONT reads corresponding to each configuration pattern.** After quality trimming (q10, headcrop 100), P2-related ONT reads longer than 10 kb were selected and classified according to configuration patterns as indicated. In the IS26 and 193-bp partial IS26 sequences indicated in light yellow, ONT reads terminated within these sequences. In the 193-bp sequence indicated in dark yellow, ONT reads probably terminated in the 193-bp sequence downstream of IS26R, but a possibility that they terminated in the 193-bp sequence in IS26L was not excluded. Short horizontal lines indicate the sequence overlaps between the two passenger sequences (PSs) at the read ends. These overlapping sequences were derived from different PS copies, not from the same copy in a monomeric, dimeric or trimeric form of cP2 because such reads are not obtained by ONT single molecule sequencing.

enzyme AvrII, which cleaves at one site in the P2 sequence (Fig. 3C). After AvrII digestion, the bands of circular DNA molecules presumably derived from mono- or multimeric cP2 molecules (open arrowheads in Fig. 3C) disappeared, and a single band appeared at the position corresponding to the size of the linearized cP2 molecule (36.5 kb).

All these data, together with the results of long-read sequence analysis, indicated that KpWEA4-2 contained not only multiple forms of P1 where the tandem amplification of pP2 occurred but also mono- and multimeric forms of cP2 molecules. Accordingly, in this manuscript, we labelled the P1 derivatives detected in this series of analyses as P1[1], P1[2] and P1[3], each of which contained one, two or three copies of pP2, respectively (Fig. 3). Similarly, the pP2-containing fragments of P1 were labelled fP1[1], fP1[2], and fP1[3], and the cP2 derivatives were labelled cP2[1], cP2[2], cP2[3], and cP2[4]. Notably, KpWEA4-2 probably contained P1- and cP2-derivatives that contained more copies of P2, but their copy numbers were below the detection levels in this series of analyses. In addition, the results shown in Figs. 2 and 3A suggest that KpWEA3 and KpWEA4-1 also contained similar sets of P2-containing molecules.

### Changes in the copy numbers of P2 during cultivation in the absence or presence of antimicrobial selection pressure

Because P2 encodes four AMR genes conferring resistance to third-generation beta-lactams, tetracyclines, sulphonamides, and trimethoprim (Fig. 1B), continuous exposure to these antimicrobials may initiate P2

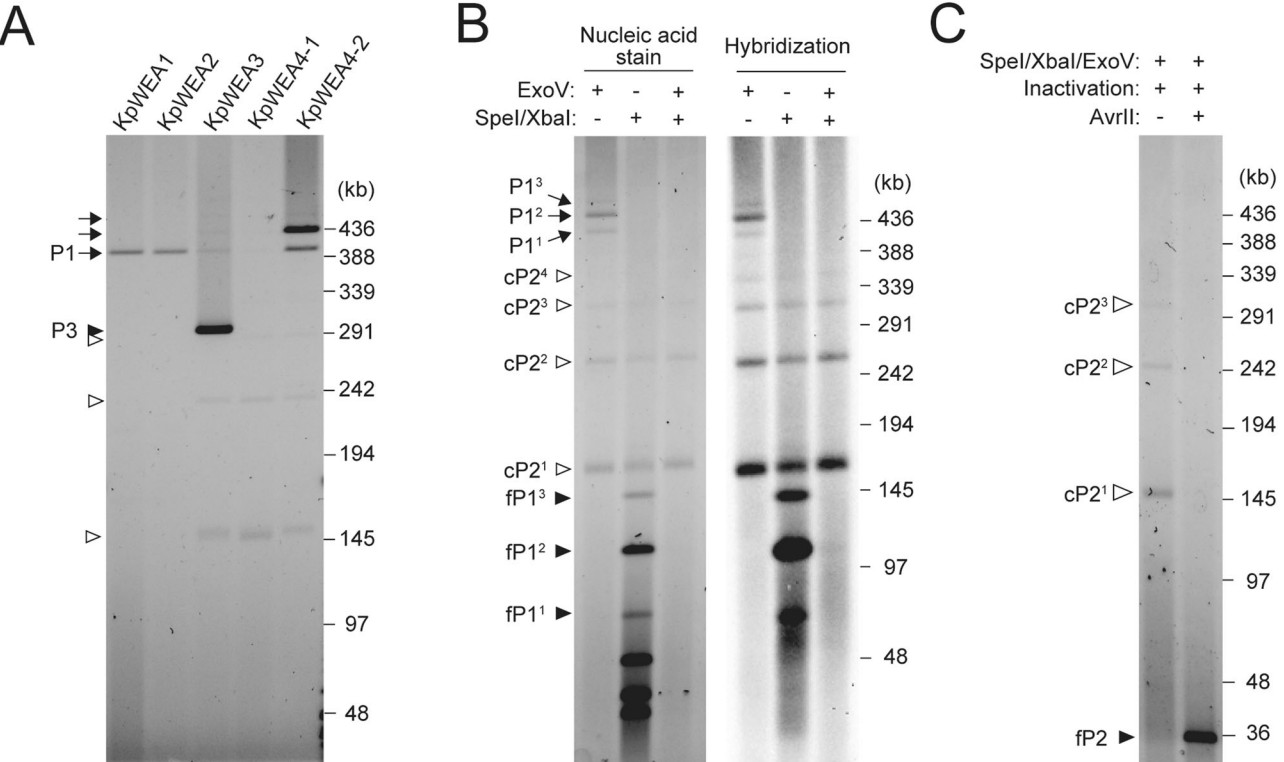

**Fig. 3 | PFGE analysis of extrachromosomal elements of the *K. pneumoniae* isolates. A** The profiles of extrachromosomal elements extracted from the five *K. pneumoniae* isolates. The P1 plasmid and its derivatives are indicated by arrows, and the P3 plasmid is indicated by a closed arrowhead. Bands indicated by open arrowheads are related to cP2 in panels (**B**) and (**C**). **B** Digestion patterns of the extrachromosomal elements of KpWEA4-2 by SpeI/XbaI and/or ExoV. P1 derivatives (P1[1], P1[2] and P1[3]) and mono- and multimeric cP2 (cP2[1], cP2[2], cP2[3] and cP2[4]) are indicated by arrows and open arrowheads, respectively. The SpeI/XbaI digestion-derived fragments of P1, which contained one, two or three copies of pP2 (fP1[1], fP1[2] and fP1[3]), are indicated by closed arrowheads. Images of a stained gel (left panel) and its blotting membrane hybridized with a P2 sequence-specific DNA probe (right panel) are shown. **C** AvrII digestion of the SpeI/XbaI/ExoV-digested extra-chromosomal elements of KpWEA4-2. As the P2 sequence contains one AvrII site, the AvrII digestion of cP2 molecules that remained undigested by SpeI/XbaI/ExoV treatment (cP2[1], cP2[2] and cP2[3], indicated by open arrowheads) generated a single 36.5-kb band (fP2, indicated by a closed arrowhead).

amplification or induce further P2 amplification. It is also possible that the copy number of P2 decreases upon long-term cultivation in the absence of antimicrobials. To examine these possibilities as well as the stability of amplified P2, we monitored the changes in the copy numbers of P2 (pP2 plus cP2) and P1 relative to the chromosome in four isolates (KpWEA1, KpWEA3, KpWEA4-1, and KpWEA4-2) during cultivation in the presence or absence of cefotaxime (CTX), a third-generation beta-lactam (Fig. 4A). As the minimum inhibitory concentration (MIC) differed among the isolates, the concentration of CTX was adjusted to the sub-MIC of each isolate, and a single colony of each isolate was inoculated into antibiotic-free or CTX-containing LB and cultivated for 10 days with a 1/200 dilution of each culture with fresh medium every day (referred to as x1 to x10 passages). After x1, x2, x5, and x10 passages, the relative copy numbers of P2 (pP2 plus cP2) and P1 in each isolate were analysed by digital PCR (Fig. 4B). In addition, after x1 and x10 passages, the cultures of KpWEA4-2 were inoculated into 50 ml of antibiotic-free or CTX-containing LB and cultured overnight to analyse the extrachromosomal DNA by PFGE (Fig. 4D). Similarly, the extrachromosomal DNA of x10-passaged KpWEA1 was also analysed by PFGE.

**Induction of P2 amplification in KpWEA1 under antimicrobial selection pressure.** As shown in Fig. 4B, in both the absence and presence of CTX, the relative copy number of P1 was constant in the four isolates and ranged from 1.1 to 2.0 throughout the passages (grey lines in Fig. 4B). In KpWEA1, while the relative copy number of P2 was unchanged until x5 passages in the presence of CTX, it increased 4-fold after x10 passages. We performed two additional experiments using other colonies of KpWEA1 and observed a similar increase in the relative

copy number of P2 after x10 passages (Supplementary Fig. 2). Furthermore, PFGE analysis of the x10-passaged KpWEA1 revealed the generation of P1[2], P1[3], cP2[1] and cP2[2] (Fig. 4C). The P1 plasmid that contained four copies of pP2 (P1[4]) was also detected. These results demonstrated that the tandem amplification of pP2 and the generation of mono- and dimeric forms of cP2 were induced in the presence of CTX. Thus, P2 amplification was reproduced in vitro in the presence of selective antimicrobial pressure. As such amplification was not detected in the absence of CTX, antimicrobial pressure was the driving force of P2 amplification.

**Stability of P2 amplification.** In KpWEA3, KpWEA4-1 and KpWEA4-2, the relative copy numbers of P2 gradually decreased through the passages in the absence of CTX (Fig. 4B). However, these were 1.9 to 4.7 times higher than that of P1 even after x10 passages. In the presence of CTX, the copy number of P2 increased in KpWEA3 (from 9.7 copies to 14.7 copies). In KpWEA4-1, the copy number first increased and then decreased, but the copy number after x10 passages was much higher than that in CTX-free medium (7.1 copies vs. 3.5 copies). In KpWEA4-2, where the highest level of P2 amplification was found, the copy number of P2 gradually decreased but was also much higher than that in CTX-free medium after x10 passages (8.5 copies vs. 4.8 copies). In the PFGE analysis of the extrachromosomal elements of KpWEA4-2 after x1 and x10 passages (Fig. 4D), P1[1], P1[2], cP2[1], and cP2[2] were detected after x10 passages even in the absence of CTX, although a P1[2]-to-P1[1] shift was observed. These results indicate that the amplified state of P2 is not fully stable but can be maintained even in the absence of selective antimicrobial pressure.

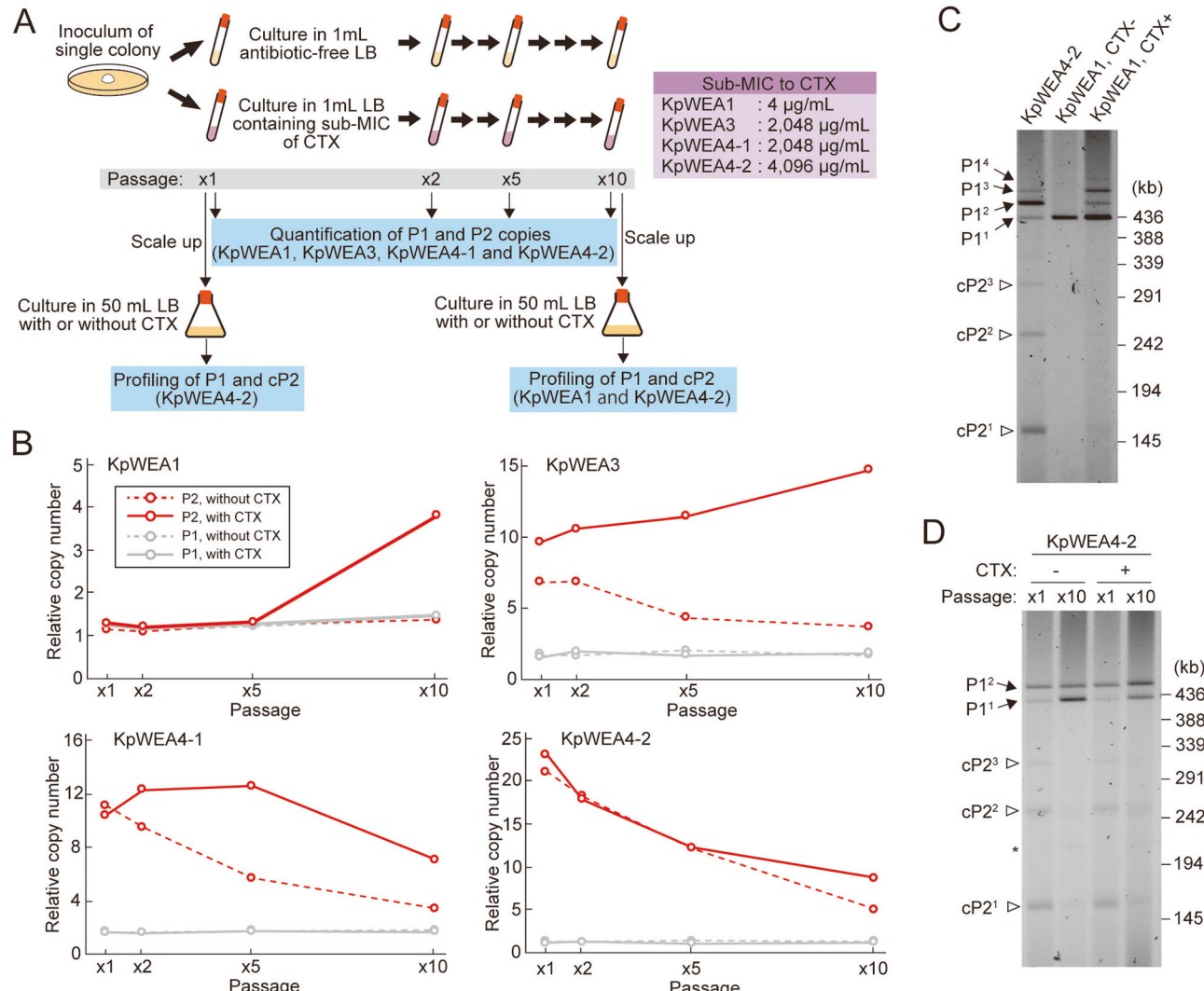

**Fig. 4 | In vitro passages of the *K. pneumoniae* isolates in the presence or absence of antimicrobial selection pressure. A** Schematic presentation of the experimental design. For each *K. pneumoniae* isolate, a single colony was subcultured 10 times (x10 passages) in the presence or absence of cefotaxime (CTX) at the sub-MICs of each isolate, and the changes in the copy numbers of the P1 and P2 sequences relative to that of the chromosome were monitored. **B** Changes in the relative copy numbers of P1-related (grey) and P2-related (red) sequences in four *K. pneumoniae* isolates. **C** The PFGE profiles of extrachromosomal elements of KpWEA1 after x10 passages in the absence or presence of CTX. The extrachromosomal elements of KpWEA4-2

shown in panel (**B**) of Fig. 3 were used as a control to show the positions of P1 derivatives (P1$^1$, P1$^2$, and P1$^3$) and cP2 (cP2$^1$, cP2$^2$ and cP2$^3$). A band that probably corresponded to P1$^4$ was also observed in KpWEA1 cultured in the presence of CTX. **D** The profiles of extrachromosomal elements of KpWEA4-2 at the x1 and x10 passages in the presence or absence of CTX. P1 derivatives (P1$^1$ and P1$^2$) and cP2 derivatives (cP2$^1$, cP2$^2$ and cP2$^3$) are indicated by arrows and open arrowheads, respectively. A faint band probably corresponding to the covalently closed circular form of P1$^1$ is indicated by an asterisk.

## Heterogeneity in the P2 amplification state in cultured bacterial populations

As the P2 amplification state showed dynamic changes during cultivation, it is likely that there is a certain level of heterogeneity in a cultured cell population. To verify this hypothesis, three colonies of KpWEA4-2 (Colonies A, B and C) were grown overnight on antibiotic-free LB plates, and a single colony was randomly selected from each plate and cultured on antibiotic-free LB plates. After repeating this procedure 10 times (x10 passages), a single colony was cultured in antibiotic-free LB overnight, and its extrachromosomal elements were analysed. Three independent series of passages (series 1 to 3) were carried out from the same initial colony for each colony (Fig. 5A). We first analysed the copy numbers of P1 and P2 relative to that of the chromosome. However, as the P1 copy numbers showed some variation, ranging from 1.3 to 1.8, the relative copy numbers of P2 were additionally normalized by that of P1 (Fig. 5B). This analysis revealed that although dynamic changes in the copy number of P2 were observed for each colony as expected, notable differences in the pattern of

copy number change were observed among colonies and among passage series. In Colony A, the copy number of P2 decreased from 20.9 to 13.5, 7.3, and 13.6 in the three series of passages. In Colony B, the copy number decreased from 18.8 to 7.9, 12.1, and 3.6. In Colony C, the copy number decreased from 22.5 to 10.6, 11.8, and 15.2. Although these copy numbers were the averages of each culture where a certain level of heterogeneity might have additionally been generated by overnight culture, this result clearly indicated the presence of heterogeneity in the P2 amplification state in each population of KpWEA4-2 cultures. PFGE analysis of the extrachromosomal elements obtained from three series of each colony before and after x10 passages also demonstrated a clear variation in the content among colonies and among passage series (Fig. 5C). Of the variations observed, the variation in P1-related molecules of Colony C was interesting. While P1$^3$ and P1$^4$ were not clearly observed in the culture of nonpassaged colonies, they were clearly detected in addition to P1$^1$ and P1$^2$ in the series 2 and 3 cultures (Fig. 5C), showing a striking contrast to series 1, where P1$^1$ became dominant.

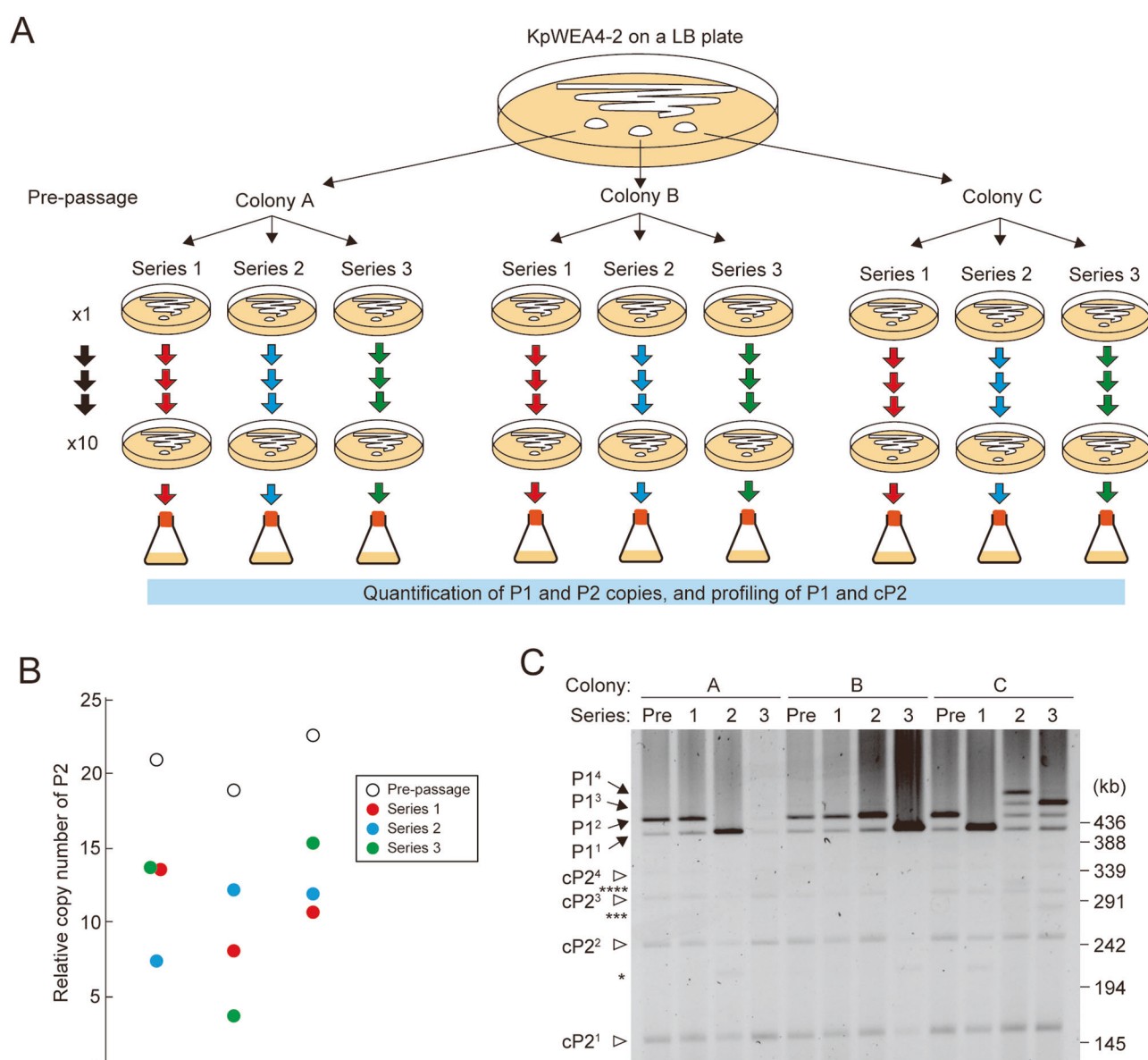

**Fig. 5 | Stability of the amplified state of the P2 sequence in the isolate KpWEA4-2 by subculturing on antibiotic-free LB plates. A** Schematic presentation of the experimental design. Three KpWEA4-2 colonies (Colonies A, B, and C) were subjected to triplicated single-colony passages (series 1, 2 and 3) on antibiotic-free LB plates, and the changes in the copy numbers of the P1 and P2 sequences relative to that of the chromosome were determined after x10 passages. **B** Copy numbers of the P2 sequence relative to that of P1 in prepassaged and passaged KpWEA4-2 cells.

**C** Changes in the PFGE profiles of extrachromosomal elements after passage. The profiles of prepassaged and passaged cells are shown. The P1 derivatives (P1$^1$, P1$^2$, P1$^3$ and P1$^4$) and the cP2 derivatives (cP2$^1$, cP2$^2$, cP2$^3$ and cP2$^4$) are indicated by arrows and open arrowheads, respectively. Faint bands indicated by single, triple and quadruple asterisks probably correspond to the covalently closed circular forms of P1$^1$, P1$^3$ and P1$^4$, respectively.

## RecA-dependent generation of cP2 and amplification of P2

P2 amplification and the generation of cP2 can be induced by RecA-dependent homologous recombination (HR) or in a RecA-independent manner, in which IS TPase is involved[14]. To identify the mechanism involved in P2 amplification and cP2 generation, we electroporated crude extrachromosomal DNA from passaged KpWEA4-2 cells that contained multiple forms of P1 and cP2 (Colony C, series 2, x10 passages in Fig. 5C) into the *recA*-deficient (Δ*recA*) *Escherichia coli* strain HST08. The transformants grown overnight on an LB plate containing 0.5 μg/mL CTX were analysed for their P2 copy numbers relative to that of P1 (Fig. 6A). Based on the data, we selected four transformants that contained approximately one, two, three, or four copies of P2 (Fig. 6A) and profiled their

extrachromosomal elements. Each transformant contained a DNA molecule corresponding to P1$^1$, P1$^2$, P1$^3$ and P1$^4$ (Fig. 6B), which was confirmed by SpeI/XbaI digestion that yielded fragments containing one, two, three or four copies of pP2 (76.5, 113, 149.5 and 186 kb, respectively) (Fig. 6C). However, cP2 was not detected in any of the transformants, as indicated by SpeI/XbaI/ExoV digestion (Fig. 6B). These results indicate that tandemly amplified pP2 can be stably maintained under the Δ*recA* genetic background and suggest that RecA-dependent HR is required for the generation of cP2.

As the four types of P1 derivatives (P1$^1$, P1$^2$, P1$^3$, and P1$^4$) could be purified from the HST08 transformants described above, we electroporated each P1 derivative into the wild-type (wt) *E. coli* strain BW25113 and its

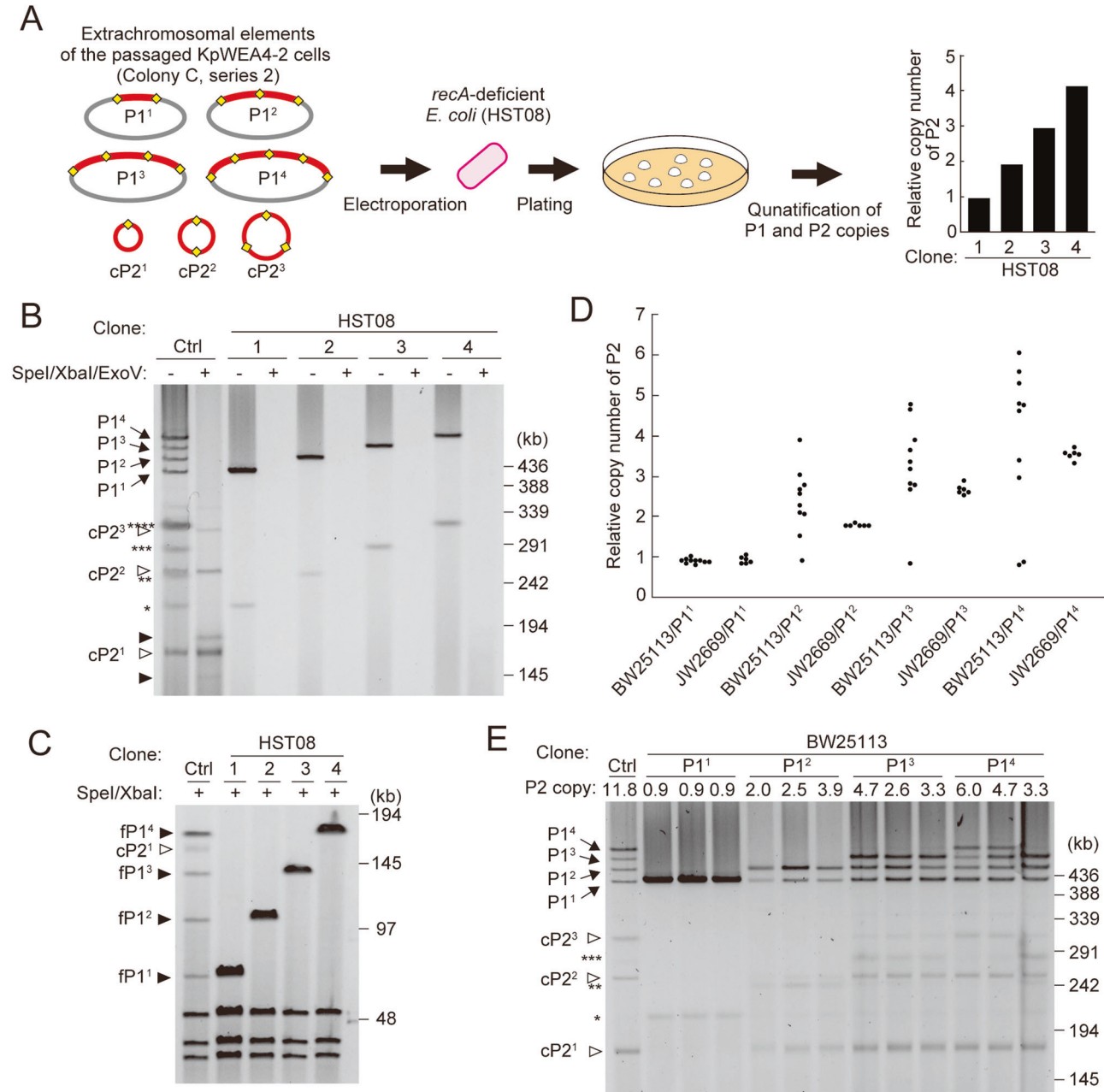

**Fig. 6 | RecA-dependent amplification of the P2 sequence in *E. coli.***
**A** Electroporation of the extrachromosomal elements obtained from the passaged KpWEA4-2 (Colony C, series 2 in Fig. 5C) into a ΔrecA *E. coli* strain (HST08). The transformants were selected on an LB plate containing 0.5 µg/mL CTX, and the copy numbers of P2 relative to that of P1 in each transformant were determined. The transformants contained approximately one, two, three, or four copies of P2. The relative copy numbers of a transformant representing each group (clones 1, 2, 3, and 4, respectively) are shown. **B** The PFGE profiles of the extrachromosomal elements of the four clones shown in panel (**A**) with or without SpeI/XbaI/ExoV treatment. The extrachromosomal elements of passaged KpWEA4-2 were used as controls (Ctrl) to show the P1 and cP2 derivatives. The single, double, triple, and quadruple asterisks indicate the covalently closed circular forms of P1¹, P1², P1³, and P1⁴, respectively, in the untreated samples. The closed arrowheads in the SpeI/XbaI/ExoV-treated KpWEA4-2 sample (Ctrl) indicate the P1³- and P1⁴-derived fragments

that were generated by incomplete digestion by ExoV. **C** SpeI/XbaI digestion of the extrachromosomal elements of the four clones shown in panel (**A**). P1-derived fragments containing one, two, three and four copies of pP2 (fP1¹, fP1², fP1³ and fP1⁴) and cP2¹ are indicated by closed arrowheads and an open arrowhead, respectively. **D** The copy numbers of P2 relative to that of P1 in *E. coli* BW25113 and its ΔrecA mutant JW2669. The P1¹, P1², P1³ or P1⁴ DNA purified from the four clones shown in panel (**A**) was introduced into the two *E. coli* strains by electroporation. Transformants were selected on LB plates containing 0.5 µg/mL CTX, and the copy numbers of P2 in ten BW25113- and six JW2669-transformants for each type of P1 derivative were examined. **E** The PFGE profiles of extrachromosomal elements of *E. coli* BW25113 transformants. For the P1²-, P1³- and P1⁴-introduced transformants, three transformants, with relatively high, intermediate, or low copy numbers of P2, in each group are shown. The extrachromosomal elements of passaged KpWEA4-2 were also used as controls (labelled Ctrl).

---

ΔrecA mutant JW2669[20]. From transformants grown overnight on LB plates containing 0.5 µg/mL CTX, ten and six transformants of the wt and ΔrecA mutant strains, respectively, were randomly selected for each P1 derivative and cultured in antibiotic-free LB to measure the copy number of P2 relative to that of P1.

All P1¹-introduced transformants either from the wt or ΔrecA mutant contained one copy of P2 (Fig. 6D). This was confirmed by PFGE analysis of three randomly selected transformants, which showed that they contained only P1¹ (Fig. 6E).

In the transformants of wt electroporated with $P1^2$, $P1^3$, or $P1^4$, notable variations in the relative copy number of P2 were observed: 0.9-3.9 for $P1^2$, 0.8-4.7 for $P1^3$, and 0.8-6.0 for $P1^4$ (Fig. 6D). However, the copy numbers of P2 were higher than those of the introduced P1 derivatives in most transformants (8, 6, and 6 out of the 10 transformants analysed for each of $P1^2$, $P1^3$, and $P1^4$, respectively), indicating that P2 was amplified in these transformants. To examine the amplification states of P2 in each group of transformants, we selected three transformants from each group so that they represented the variation in the copy number of P2 in each group (low, intermediate, and high copy numbers) and profiled their extrachromosomal elements (Fig. 6E). In the $P1^2$-introduced transformants, although $P1^1$ was detected in addition to $P1^2$, $cP1^1$ was detected in all transformants. Similarly, in the $P1^3$-introduced transformants, although $P1^1$ and $P1^2$ were detected in addition to $P1^3$, $cP1^1$ and $cP1^2$ were clearly detected in all transformants. In the $P1^4$-introduced transformants, $P1^1$, $P1^2$, $P1^3$ and $P1^4$ were detected in two transformants, although $P1^4$ was not detected in one transformant. Moreover, $cP1^1$, $cP1^2$ and $cP1^3$ were detected in all transformants. Thus, while further amplification of pP2 was not detected, mono- and multimeric cP2 molecules were generated under the wt genetic background. As a clear correlation between the PFGE profile and the copy number of P2 in each transformant was not observed, we examined whether the difference in the copy number of P2 among transformants showed some correlation with the difference in the MICs to CTX (Supplementary Table 2). This analysis revealed that the MIC was elevated to 128 µg/mL by the introduction of $P1^2$, 4- to 8-fold higher than the MIC of the $P1^1$-introduced transformants, but no difference was detected among the $P1^2$-, $P1^3$- and $P1^4$-introduced transformants. This was probably because the MIC of $P1^2$-introduced transformants was too high to detect further differences.

In sharp contrast to that in the transformants of wt, the copy numbers of P2 in all transformants of the $\Delta recA$ mutant were nearly the same as those of the introduced P1 derivatives, and there was no clear increase or variation in the P2 copy number (Fig. 6D). This result is consistent with the result of the analysis using E. coli HST08 (Fig. 6A, B).

The findings obtained from this series of analyses indicate that RecA-dependent HR plays a critical role in P2 amplification and the generation of mono- and multimeric forms of cP2.

## Analysis of P2 amplification using a mini-PCTn and its TPase-deficient derivative

As described above, RecA-dependent HR plays a critical role in P2 amplification and cP2 generation (Fig. 6). However, it is unknown whether IS26 TPase is involved in these genetic events. To investigate it, we constructed a "mini-PCTn" on pSY396, an 8.5-kb single-copy plasmid, to generate pMA026 (Fig. 7A). The mini-PCTn was approximately 3 kb in length and contained only the $bla_{CTX-M-14}$ gene of the original PS (35.5 kb). Thus, the 2.2-kb sequence comprising IS26L and the $bla_{CTX-M-14}$-encoding sequence corresponded to the original pP2 (referred to as "mini-pP2"; Fig. 7A). As two BamHI sites flanked the mini-PCTn, the length of the mini-PCTn in pMA026 can be monitored by BamHI digestion. We also constructed a pMA026-derivative named pMA053 (Fig. 7A), in which both TPases encoded by IS26L and IS26R were inactivated by inserting a 4-bp sequence (c.55_56insGTAC). This mutation was previously shown to abolish the TPase-dependent TU formation[14]. We introduced each plasmid into E. coli BW25113 (wt) and JW2669 ($\Delta recA$ mutant) by electroporation, and transformants were selected on LB plates containing 0.5 µg/mL CTX. Then, a single colony of each transformant was inoculated into LB containing 0.5 or 16 µg/mL CTX and subjected to three independent series of x5 passages similar to that illustrated in Fig. 4A, and the average copy numbers of the $bla_{CTX-M-14}$ gene relative to those of the plasmid-encoded sopA gene were determined by digital PCR. Profiles of BamHI-digested extrachromosomal DNA were also analysed using field-inversion gel electrophoresis (FIGE).

When pMA026-introduced BW25113 was cultured (x5 passages) in the presence of 0.5 µg/mL CTX, the relative copy number of $bla_{CTX-M-14}$ was approximately 1 (Fig. 7B). BamHI digestion of the extrachromosomal DNA yielded two fragments of approximately 3 and 8.5 kb in length,

corresponding to the mini-PCTn and a plasmid vector, respectively (Fig. 7C). In contrast, by culturing in the presence of 16 µg/mL CTX, the relative copy number of $bla_{CTX-M-14}$ was significantly increased (10.0 on average, Fig. 7B). In addition, BamHI digestion of the extrachromosomal DNA yielded a fragment of approximately 22.5 kb in length instead of the 3-kb fragment (Fig. 7C). These results indicated that the 2.2-kb mini-pP2 was repeated 10 times (x10 amplification) in pMA026 under antimicrobial selection pressure.

In the pMA026-introduced BW25113 cells cultured in the presence of 16 µg/mL CTX, circular molecules corresponding to cP2 (referred to as "mini-cP2") were not detected when the same amount of extrachromosomal DNA as that used in Fig. 7C (0.1 µg) was treated with BamHI and ExoV (Supplementary Fig. 3). As this failure appeared to be due to the lower molecular weight of mini-cP2 and/or the presence of various forms of multimeric mini-cP2, we attempted to detect a 2.2-kb fragment derived from mini-cP2 molecules by using an increased amount of DNA and modifying the protocol. In this analysis, 1 µg extrachromosomal DNA reextracted from the pMA026-introduced BW25113 cells cultured in the presence of 16 µg/mL CTX (the sample from the same series as that shown in Fig. 7C) was treated with NdeI, SpeI and XhoI to digest the plasmid vector sequence and with ExoV to remove linear DNA molecules, and then digested with PmeI. As PmeI cleaves the 2.2-kb sequence at one site, mono- and multimeric forms of mini-cP2 can be converted to a 2.2-kb fragment. By this procedure, we could detect a 2.2-kb fragment (Fig. 7D), confirming the generation of mini-cP2. Minor bands most probably derived from the mini-PCTn, in which x11 or less than x10 amplification of mini-pP2 occurred, were also detected in this analysis. Thus, this experimental system reproduced the tandem amplification of pP2 and the generation of cP2 observed for the original plasmid.

When pMA026-introduced JW2669 was cultured in the presence of 16 µg/mL CTX, the relative copy number of $bla_{CTX-M-14}$ was approximately 2 (Fig. 7B), much lower than that observed in the wt E. coli strain (BW25113). However, BamHI digestion of the extrachromosomal DNA yielded two fragments corresponding to the mini-PCTn containing x2 and x4 amplified mini-pP2, respectively (Fig. 7C). These results indicated that the amplification of mini-pP2 can occur in the RecA-deficient genetic background under antimicrobial selection pressure, although the efficiency of amplification was much lower than in the RecA-active genetic background.

When pMA053-introduced BW25113 was cultured in the presence of 16 µg/mL CTX, the copy number of $bla_{CTX-M-14}$ was increased, albeit to a lesser extent compared with that of pMA026 in BW25113 (5.9 vs. 10.0, Fig. 7B). BamHI digestion of the extrachromosomal DNA from the pMA053-introduced BW25113 cells cultured in the presence of 16 µg/mL CTX also yielded several fragments derived from the mini-PCTn, in which x6, x7, x9, or x11 amplification of mini-pP2 occurred (Fig. 7C). These results indicate that the tandem amplification of mini-pP2 can occur in the absence of active TPase although amplification efficacy was lower than that in the cells where active TPase was present.

Unexpectedly but intriguingly, when pMA053-introduced JW2669 was cultured in the presence of 16 µg/mL CTX, although the relative copy of $bla_{CTX-M-14}$ was only slightly increased (approximately 1.5 on average, Fig. 7B), BamHI digestion of the extrachromosomal DNA yielded a fragment which was apparently derived from the mini-PCTn, in which x2 amplification of mini-pP2 occurred (Fig. 7C). This unexpected finding indicates the presence of some mechanism(s) that generates the tandem amplification of mini-pP2 in the absence of active RecA and TPase.

Collectively, although the results of this series of experiments using plasmids pMA026 and pMA053 support the notion that RecA-dependent HR plays a major role in the tandem amplification of pP2 in the presence of antimicrobial selective pressure, the tandem amplification of mini-pP2 can occur in the absence of RecA. Moreover, although it is still not clear whether and how TPase is involved in this process, there is an unknown RecA/TPase-independent mechanism(s) that generates the tandem amplification of mini-pP2.

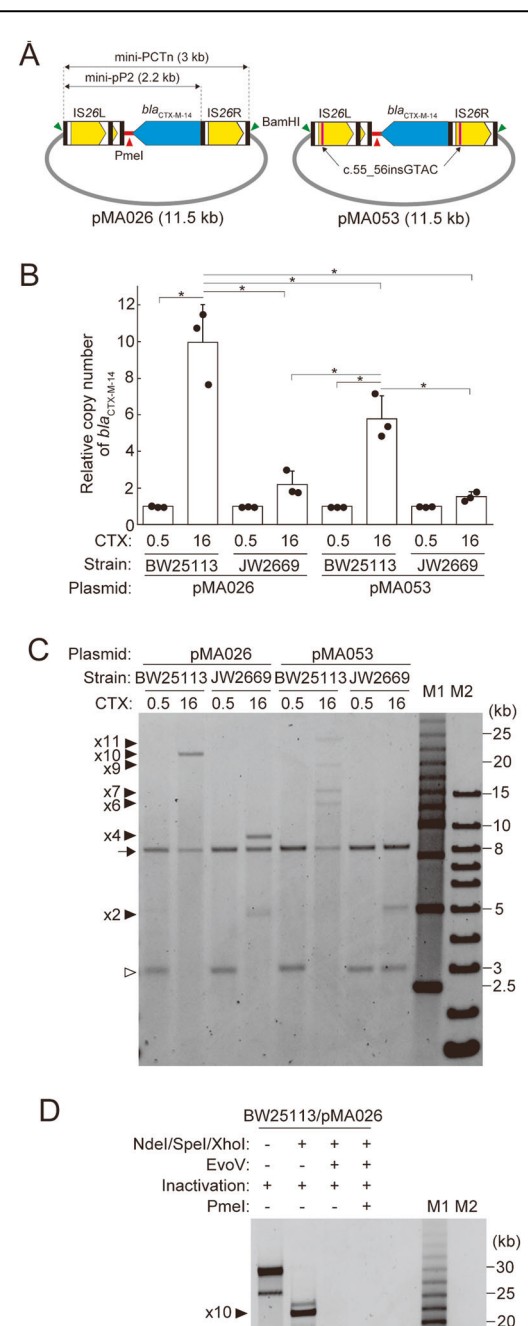

**Fig. 7 | Construction of pSY396-derived, single copy plasmids carrying the mini-PCTn or its TPase-deficient derivative and analyses of the amplification of mini-pP2 in the wild-type *E. coli* and its RecA-deficient mutant. A** The structures of plasmids carrying the mini-PCTn (pMA026) and its TPase-deficient derivative (pMA053). The mini-PCTn contained the $bla_{CTX-M-14}$ gene instead of the 35.5-kb PS in the original PCTn. The both TPases in IS26L and IS26R were inactivated by inserting a 4-bp (c.55_56insGTAC) in pMA053. The positions of BamHI and PmeI sites are indicated by green and red arrowheads, respectively. **B** The average copy numbers of $bla_{CTX-M-14}$ gene relative to those of the plasmid-encoded *sopA* gene in *E. coli* BW25113 and its ΔrecA mutant JW2669 after x5 passages in the absence or presence of antimicrobial selection pressure. pMA026- or pMA053-introduced BW25113 and JW2669 were cultured in 0.5 or 16 μg/mL CTX-containing LB. After x5 passages, the copy numbers of $bla_{CTX-M-14}$ and *sopA* were determined by digital PCR. The values represent the mean ± standard deviation (s.d.) from three independent experiments. Statistical analyses were performed by one-way ANOVA with Tukey-Kramer HSD test. The asterisk indicates *p* value < 0.05. **C** The BamHI-digestion profiles of pMA026, pMA053 and their derivatives in the BW25113 and JW2669 cells shown in panel (**B**). The arrow and open arrowhead indicate fragments derived from the vector plasmid sequence and the mini-PCTn where no amplification of mini-pP2 occurred, respectively. Filled arrowheads indicate fragments derived from the mini-PCTn where the mini-pP2 was repeated 2, 4, 6, 7, 9, 10, and 11 times (x2, x4, x6, x7, x9, x10, and x11). M1 and M2 indicate the 2.5 kb molecular ruler (Bio-Rad) and 1 Kb Plus DNA ladder (Thermo Fisher Scientific), respectively. **D** Detection of a fragment derived from the mini-cP2 in the pMA026-introduced BW25113 cells using a modified protocol. The extrachromosomal DNA (1 μg) extracted from the pMA026-introduced BW25113 cells cultured in the presence of 16 μg/mL CTX were digested with NedI, SpeI and XhoI in the presence or absence of ExoV. After heat inactivation of the enzymes, residual DNA molecules were digested with PmeI to convert mono-and multimeric mini-cP2 into a 2.2-kb fragment (indicated by the open arrowhead). Arrows indicate fragments derived from the vector plasmid sequence after NdeI/SpeI/XhoI-digestion. The filled arrowhead indicates the fragment derived from the mini-PCTn where mini-pP2 was repeated x10. Note that minor fragments probably derived from the mini-PCTn where mini-pP2 repeated x11 or less than x10 were also detected by analysing an increased amount of extrachromosomal DNA (1 μg).

preexisting plasmid, P1 (Fig. 1B). Through several series of analyses, we obtained multiple lines of evidence important for understanding TU-mediated genome amplification.

First, by analysing the copy number of the P2 sequence, long-read sequences and PFGE-based profiles of extrachromosomal elements of the original *K. pneumoniae* isolates, we revealed that the P2 sequence in the P1 plasmid (pP2) was tandemly amplified and that not only mono- but also multimeric forms of cP2 were present (Figs. 2 and 3). Although tandem amplification of TU has been described in several MDR bacteria[8–12,21,22], the presence of excised and circularized TU was detected only by PCR for a TU derived from Tn4532B, a derivative of Tn4532[14,16]. Therefore, to our knowledge, this is the first visual detection of a circular form of TU. Moreover, the presence of multimeric forms of circular TU has not been described in any bacteria thus far. It is currently unknown why a large amount of circularized TU was generated in the present case. One possibility is that the presence of a partial IS26 copy directly next to the end of IS26L contributed to the active generation of cP2 because an insertion of two bases at the end of IS26 in Tn4532B affected the efficiency of TU formation[14,16], but this needs to be verified experimentally. Similar to the present case, Tn6020, which is amplified in *Acinetobacter baumannii* strains, included a 3'-terminal sequence (175 bp) of IS26 next to the left end of intact IS26[8,23].

Second, by subculturing the *K. pneumoniae* isolates, we found that the amplified states of P2 were maintained with notable stability even in the absence of antimicrobials. This is an unexpected result because, in general, a high level of genome amplification is costly and disadvantageous for bacterial cells in the absence of specific selection pressures[24–26]. In fact, in a clinical *E. coli* isolate in which a 47-kb IS26-associated MDR region was tandemly amplified, the amplification was immediately resolved by culturing in antibiotic-free medium[9]. In this regard, the comparison of growth rates among the subclones of *K. pneumoniae* isolate obtained in this study may be informative, but this comparison is difficult because the

## Discussion

In our previous study analysing a *K. pneumoniae* MDR clone[19], we found that the clone acquired a strange circular extrachromosomal element during within-host evolution (Fig. 1A). This element, named P2, was unique in that it contained no replicon but was inherited by descendant subclones. Here, we show that this element (referred to as cP2 in this manuscript) is a circular form of a TU derived from an IS26-associated PCTn inserted in a

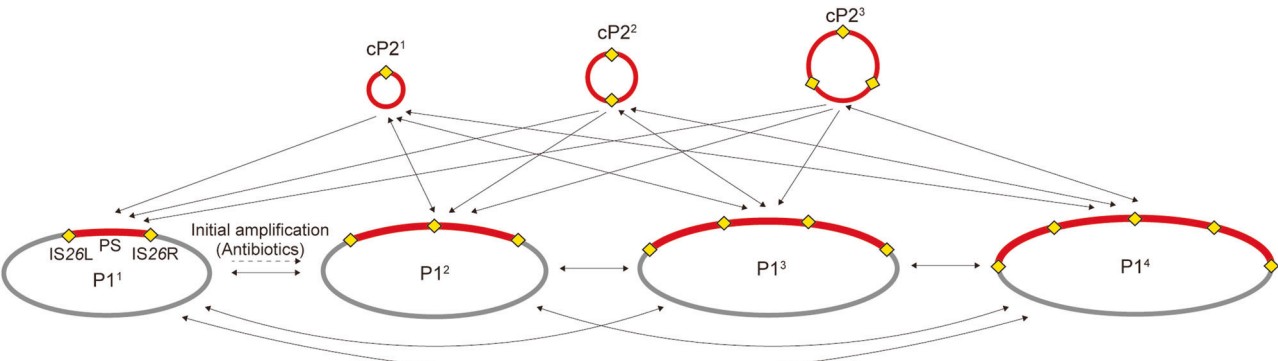

**Fig. 8 | Possible pathways for the amplification of the P2 sequence.** The amplification of pP2 is generated by a RecA-dependent or independent manner in the presence of antimicrobial pressure. The following amplification of pP2 and the generation of mono- and multimeric forms of cP2 were induced by RecA-dependent homologous recombination. Although the amplified state of the P2 sequence is highly dynamic, as the deletion of the P2 sequence is also induced by RecA-dependent recombination, the amplified state of the P2 sequence can be maintained at a certain level once the P2 sequence has been highly amplified by the two modes of amplification (tandem amplification of pP2 in the P1 plasmid and the generation of mono- and multimeric cP2).

amplification states of P2 are highly variable and can probably generate some variation even after overnight cultivation. However, it is noteworthy that *E. coli* transformants stably maintaining two, three or four copies of pP2 did not show a clear difference in growth rate compared with that containing a single copy of pP2 (Supplementary Fig. 4). There may be some specific mechanism(s) allowing the relatively stable maintenance of the amplified state of P2, but this mechanism is currently unknown.

Third, by in vitro passage of a *K. pneumoniae* isolate, in which P2 amplification had not yet occurred (KpWEA1), in the presence of the sub-MIC of a relevant antimicrobial (CTX), the generation of tandemly amplified pP2 and mono- and multimeric cP2 were reproduced (Fig. 4 and Supplementary Fig. 2). Moreover, by the in vitro passage of *K. pneumoniae* isolates, in which P2 amplification had already occurred, in the presence of CTX, P2 was further amplified by two modes of amplification, namely, tandem amplification of pP2 and the generation of multimeric cP2 (Fig. 4). These results clearly indicate that antimicrobial selection pressure is a driving force of the two modes of P2 amplification. This can explain why other regions in the P1 plasmid, which potentially form IS*26*-associated TU structures (Supplementary Fig. 1), were not amplified during the within-host evolution of this *K. pneumoniae* clone.

Finally, by using Δ*recA E. coli* strains, we showed that RecA function is required for the formation of mono- and multimeric forms of cP2 in strains containing tandemly amplified pP2 (Fig. 6). Although we could not exclude the possibility that a small portion of these cP2 molecules was generated in an IS*26* TPase-dependent manner[14], this result indicates that RecA-dependent HR mainly contributes to the formation of cP2. Moreover, by using plasmids carrying the mini-PCTn (pMA026) and its TPase-deficient derivative (pMA053), we showed that although it is still not clear whether and how TPase is involved in this process, the tandem amplification of mini-pP2 can occur in the RecA/TPase-independent manner under antimicrobial selection pressure, albeit at a low efficiency (Fig. 7).

Based on the findings of the study, we propose a mechanism for the amplification and maintenance of P2 as shown in Fig. 8. At first, the initial amplification of pP2 in the P1 plasmid occurred under antimicrobial selection pressure. Both RecA-dependent and independent mechanisms could be involved in this step. Once the tandem amplification of pP2 occurred (the formation of P1²), RecA-dependent HR generated a monomeric form of cP2 from P1², which could be recombined to the pP2 region in the replicating P1 genome to generate a P1³ plasmid. By the same mechanism, P1 derivatives containing more copies of pP2 and multimeric forms of cP2 could be generated. As RecA-dependent HR could also induce the deletion of the amplified P2 sequence, these bidirectional reactions generated a highly dynamic amplification state of the P2 sequence in the population of each culture, which was not fully addressed in this study due to technical difficulties. However, once a high level of amplification was established, a certain level of stability in the cell population could be maintained. The remaining issues to be addressed in this model are whether and how TPase is involved in this process and what is the RecA/TPase-independent mechanism to generate the tandem amplification of P2. To address these issues, we need to identify the RecA/TPase-independent mechanism and examine whether the tandem amplification of pP2 (or mini-pP2) and its TPase-deficient derivative can occur under the genetic background where RecA is inactive and the putative RecA/TPase-independent mechanism does not work.

## Materials and methods
### Bacterial strains and cultivation
The clinical course and the intervals of isolation of the five *K. pneumoniae* isolates (KpWEA1, KpWEA2, KpWEA3, KpWEA4-1 and KpWEA4-2) were previously described[19]. *E. coli* HST08 competent cells (Takara Bio) were used for the transformation of the extrachromosomal DNA extracted from the passaged KpWEA4-2 cells, and the transformants were used to isolate P1¹, P1², P1³ and P1⁴ DNA. *E. coli* BW25113 and its Δ*recA* mutant JW2669 were provided by the National BioResource Project (NIG, Japan): E. coli. The *K. pneumoniae* isolates and *E. coli* strains stocked at –80 °C were grown on LB plates, and single randomly selected colonies were cultured with shaking at 37 °C in appropriate media for the following experiments. For the determination of the copy numbers of each element, colonies were cultured in 1 mL of antibiotic-free LB with shaking to the stationary phase. For the profiling of extrachromosomal elements, colonies were precultured in 1 mL of antibiotic-free LB for 5 h. Each preculture was inoculated into 50 mL of fresh LB and further cultivated to the stationary phase. To monitor the changes in the copy numbers of the P1 plasmid and P2 sequence, the same single colonies of KpWEA1, KpWEA3, KpWEA4-1, and KpWEA4-2 were respectively inoculated into 1 mL of LB containing no antimicrobial or sub-MIC CTX and cultured for 24 h. The cultures were diluted 1:200 with fresh medium and cultured again under the same conditions. This passage was repeated 10 times. Aliquots (500 μL) of the culture were collected from x10-passaged KpWEA1 and x1- or x10-passaged KpWEA4-2, inoculated into 50 mL of fresh antibiotic-free or CTX-containing LB according to the passage conditions and cultured to stationary phase to extract extrachromosomal DNA. For the analysis of the heterogeneity in the amplification state of P2, three independent colonies of KpWEA4-2 (Colonies A, B, and C in Fig. 5) were randomly selected and cultured on antibiotic-free LB plates for 24 h. Three colonies grown on each plate were randomly selected and cultured again on antibiotic-free LB plates for 24 h, and then one colony grown on each plate was selected and cultured again on antibiotic-free LB plates. This procedure was repeated 10 times (three independent series of passages for Colonies A, B, and C). After x10 passages, single colonies were cultured in 50 mL of antibiotic-free LB overnight to quantify the relative

copy numbers of P2 and profile extrachromosomal elements. To monitor the changes in copy number of the $bla_{CTX-M-14}$ gene in pMA026- or pMA053-introduced BW25113 and JW2669, their single colonies were cultured in 1 mL of LB containing 0.5 or 16 µg/mL of CTX for 24 h and subjected to x5 passages as described for *K. pneumoniae* isolates.

### Construction of pMA026 and pMA053

The outline of the processes to construct pMA026 and pMA053 is shown in Supplementary Fig. 5. Briefly, we synthesised a DNA fragment that encoded a BamHI site, IS*26L*, a partial (193 bp) IS*26*, and multiple cloning sites, and inserted this fragment and the $bla_{CTX-M-14}$ gene-encoding fragment in the PS (positions 91,505-92,604 in accession no. AP024569) into pUC57 to generated pMA016. A 4-bp insertion (c.55_56ins.GTAC) in IS*26L* was introduced by site-directed mutagenesis using the primers shown in Supplementary Table 1 to generate pMA024. Then, the intact and mutated IS*26L* sequences were amplified using primers designed to add a BamHI site to the 3'-terminus of amplicons (Supplementary Table 1). Then, by using NEBuilder HiFi DNA Assembly (New England Biolabs), the intact or mutated IS*26R* amplicons were inserted next to the $bla_{CTX-M14}$ gene in pMA016 and pMA024. Finally, the elements flanking by intact or mutated IS*26* were excised from pMA017 and pMA034, respectively, by BamHI digestion and inserted into a BamHI site of pSY396, a single-copy plasmid derived from F plasmid[27], which was provided by the National BioResource Project (NIG, Japan): E. coli.

### Digital PCR

Total genomic DNA was extracted from the cultured cells using NucleoSpin Microbial DNA (Takara Bio) according to the manufacturer's instructions. The concentration of extracted DNA was adjusted to 0.04 ng/µL, and copy numbers of the P1 plasmid, the P2 sequence, the *sopA* gene in pSY396, the $bla_{CTX-M-14}$ gene, and chromosome in each DNA preparation were quantified using a QX200 Droplet Digital PCR system (Bio-Rad). To separate the tandemly amplified mini-pP2 in pMA026 and pMA053 into single copies, a restriction enzyme ApaL1 (Takara Bio), which cleaves at one site within the IS*26* sequence, was added into PCR cocktails. The primers and probes are listed in Supplementary Table 1. Since the KpWEA4-2 chromosome also contained a P2-derived sequence (a 29.5-kb sequence of P2), for the specific amplification of the P2 sequence in pP2 and cP2, the P2 amplification primers were designed outside the 29.5-kb sequence.

### Configuration pattern analysis of P2-related ONT reads

The complete genome sequences of each *K. pneumoniae* isolate were determined in our previous study by hybrid assembly of Illumina short reads and ONT long reads[19] (accession numbers: GCA_021655435 for KpWEA1, GCA_021655455 for KpWEA2, GCA_021655475 for KpWEA3, GCA_021655495 for KpWEA4-1, GCA_021655515 for KpWEA4-2). From the long reads obtained in that study (accession numbers: DRX477151 for KpWEA1, DRX477152 for KpWEA2, DRX477153 for KpWEA3, DRX477154 for KpWEA4-1, DRX477155 for KpWEA4-2), >10 kb reads were selected and mapped to the complete genomes of each isolate using minimap2 v 2.24[28]. In the mapping for KpWEA3, KpWEA4-1 and KpWEA4-2, the cP2 sequence was excluded from the reference complete genomes. Among the mapped reads, we selected those mapped to the PS in P1 and determined whether each read included IS*26*, a partial IS*26* copy (193 bp) next to IS*26L*, PS, the 1-kb upstream region of IS*26L*, or the 1-kb downstream region of IS*26R* by Blastn analysis (v2.10.1, e-value cut-off; 1e-05) using each sequence as a query.

### Agarose gel electrophoresis

P1 and P2 derivatives were extracted from cultured cells by NucleoBond Xtra (Takara Bio) according to the manufacturer's instructions. After preparing a 1% (w/v) agarose gel with pulsed-field certified agarose (Bio-Rad), 400 ng of extracted DNA was separated using the CHEF Mapper XA system (Bio-Rad) at 14 °C by applying 6 V/cm$^2$ for 20 h. To identify bands derived from the P1 plasmid and cP2, 400 ng of DNA was digested with SpeI

(Takara Bio) and XbaI (New England Biolabs) in the presence or absence of ExoV (New England Biolabs) at 37 °C for 2 h. For the additional digestion with AvrII, SpeI/XbaI/ExoV-digested samples were incubated at 80 °C for 20 min to inactivate SpeI/XbaI/ExoV and incubated with AvrII (New England Biolabs) at 37 °C for 1 h.

To profile pMA026, pMA053 and their derivatives, extrachromosomal DNA was extracted from relevant cultured cells by NucleoSpin Plasmid EasyPure (Takara Bio) according to the manufacturer's instructions. Extracted DNA (100 ng) was digested with BamHI-HF (New England Biolabs) in the presence or absence of ExoV (New England Biolabs) at 37 °C for 2 h (Fig. 7C and Supplementary Fig. 3). To detect the 2.2-kb DNA fragment derived from mini-cP2 (Fig. 7D), the pMA026-introduced BW25113 cells, which were x5 passaged and kept in a –80 °C freezer, were cultured in 50 mL LB containing 16 µg/mL of CTX, and its extra-chromosomal DNA was extracted by NucleoBond Xtra (Takara Bio) according to the manufacturer's instructions. One µg of re-extracted DNA was digested with NdeI (New England Biolabs), SpeI (Takara Bio) and XhoI (Takara Bio) in the presence or absence of ExoV (New England Biolabs) at 37 °C for 2 h. After inactivating the enzymes at 85 °C for 20 min, undigested elements were cleaved with PmeI (New England Biolabs) at 37 °C for 1 h and subjected to FIGE using a 1% (w/v) agarose gel with pulsed-field certified agarose (Bio-Rad) and the CHEF Mapper XA system (Bio-Rad) at 14 °C by applying 9 V/cm$^2$ (forward direction) and 6 V/cm$^2$ (reverse direction) for 13.8 h. The initial and final inversion intervals were 0.05 and 0.92 seconds, respectively.

After electrophoresis, the gels were stained with SYBR Safe DNA Gel Stain (Thermo Fisher Scientific), and the images were obtained using the iBright FL1500 Imaging System (Thermo Fisher Scientific).

### Southern blotting and hybridization analysis

After PFGE, the agarose gel was incubated in 0.25 N hydrochloride for 10 min. After washing twice with distilled water, the gel was incubated in the denaturing solution (1.5 M sodium chloride and 0.5 M sodium hydroxide) for 20 min, followed by a 5-min incubation in the neutralizing solution (1.5 M sodium chloride and 1 M Tris-hydrochloride; pH 7.4). The DNA in the gel was transferred onto Hybond-N+ (Cytiva) via upwards capillary transfer and immobilized by ultraviolet crosslinking and baking. To prepare the DNA probe for hybridization, the $bla_{CTX-M-14}$ gene was amplified using the primers listed in Supplementary Table 1, and the amplified DNA fragment was purified using a QIAquick PCR purification kit (Qiagen) according to the manufacturer's instructions. Then, the fragment was labelled using Alkphos Direct Labelling Reagents (Cytiva) according to the manufacturer's instructions. Hybridization was performed at 55 °C for 24 h in hybridization buffer (Cytiva) within a glass bottle using the hybridization incubator HB-100N (Taitec). After washing the membrane with the washing buffers prepared according to the manufacturer's instructions for the labelling kit, the membrane was incubated with CDP-Star Detection Reagents (Cytiva), and the image was obtained using the iBright FL1500 Imaging System (Thermo Fisher Scientific).

### Electroporation

Competent cells of BW25113 and JW2669 were prepared according to a previously published method[29] with minor modifications[30]. The purified P1 DNA (100 ng for each of P1[1], P1[2], P1[3] and P1[4]), pMA026 or pMA053, and 100 µL of competent cell solution were mixed, transferred to a chilled 2-mm cuvette, and subjected to electroporation at 2.5 kV/cm, 25 µF and 300 Ω using the Gene Pulser II system (Bio-Rad). Immediately, 1 mL of LB supplemented with 5 mmol/L magnesium chloride was added to the cell suspension, followed by a 1-h incubation at 37 °C. The cells were inoculated onto LB plates containing 0.5 µg/mL CTX and incubated at 37 °C overnight to obtain transformants.

### Statistics and reproducibility

The differences in the relative copy number of $bla_{CTX-M-14}$ between the pMA026- and pMA053-introduced *E. coli* cells were measured by three

independent experiments and statistically analysed using one-way ANOVA with Tukey-Kramer HSD test. A $p$-value $< 0.05$ was considered to indicate statistical significance.

## Data availability

The uncropped PFGE images are shown in Supplementary Figs. 6–9. Other source data is available on Figshare (https://doi.org/10.6084/m9.figshare.24099861.v3). All ONT reads and complete genomes of the *K. pneumoniae* isolates are available in DDBJ/EMBL/GenBank under the BioProject accession number PRJDB9509.

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

## Acknowledgements

This work was supported by the Japan Society for the Promotion of Science (JSPS) KAKENHI Grant Number JP20H03579, Takeda Science Foundation, and MSD Life Science Foundation, Public Interest Incorporated Foundation. We thank our colleagues at the Department of Clinical Chemistry and Laboratory Medicine, Kyushu University Hospital, and Yuriko Sato at the Department of Bacteriology, Graduate School of Medical Sciences, Kyushu University, for their technical support.

## Author contributions

M.A., Y.G. and T.H. conceptualized the study, designed the experiments, and wrote the manuscript. M.A., S.S., Y.M., T.U. and D.K. contributed to obtaining and analysing isolates through experiments, except for genome analysis. Y.G. and T.H. contributed to genome analysis.

## Competing interests

The authors declare no competing interests.

## Ethics

This study was approved by the Centre for Clinical and Translational Research of Kyushu University Hospital (reference number 30-143).
