## [Peer review file · Communications Biology]

Reviewers' comments:

Reviewer #1 (Remarks to the Author):

This article describes the dynamics of genome amplification processes in bacteria, specifically focusing on the role of IS26 in the formation of pseudocompound transposons (PCTn) in multidrug-resistant regions of gram-negative bacteria. The study delves into the details of how these transposons can be tandemly amplified and presents evidence supporting the generation of a circular intermediate known as the "translocatable unit (TU)". The work is set within the context of *Klebsiella pneumoniae* multidrug-resistant clones.

The text, in general, is well-written, but a thorough proofreading is recommended to eliminate any minor grammatical issues.

The M&M section is well-structured and provides a comprehensive overview of the experimental procedures. Researchers familiar with the field should find it straightforward to reproduce the experiments based on this description. Indeed, the use of various techniques, including PCR, long-read sequencing, and pulsed-field gel electrophoresis (PFGE), enhances the credibility of the findings and shows a comprehensive approach to the study.

Moreover, the article describes a real-world scenario where *Klebsiella pneumoniae* isolates were obtained from a patient undergoing heavy antimicrobial treatment. This context enriches the findings by situating them within an applicable healthcare framework.

However, this manuscript has a lack of novelty in key findings. This work primarily focuses on the circular intermediate and tandem amplifications and both of these phenomena have already been described in previous studies, thus, the contribution of the present manuscript to the field might be limited. For instance, this manuscript did not provide direct evidence or demonstration of the transposition of the PCTn. Transposition is a crucial molecular mechanism that underlies the mobility of genetic elements, and its direct demonstration could have added significant value to the study. The absence of this evidence might be a potential weakness, especially if the title or the objectives of the study imply a focus on understanding the movement of the PCTn.

The authors presents a proposed model (Figure 6) that offers a coherent mechanism for the amplification and maintenance of P2, emphasizing the roles of antimicrobial pressure and RecA-dependent homologous recombination. However, there are gaps and uncertainties in the model, such as the exact mechanism of initial amplification and the potential role of IS26-associated TUs. These gaps can serve as directions for future research.

If the manuscript aims for a top-tier journal, addressing these gaps and uncertainties, even if only theoretically, might strengthen the paper. If experimental validation isn't possible in the current study, a comprehensive discussion of the implications of these gaps and their potential impacts on the model would be beneficial.

In conclusion, despite this weakness, this study provides a new contribution to the understanding of genome amplification in bacteria, especially in the context of antimicrobial resistance. The findings

presented are well-supported by rigorous methods, and they open new avenues for further exploration in the mechanisms related with IS26 transposition that might be untangled in further studies.

Reviewer #2 (Remarks to the Author):

Herein the authors build upon a 2021 MSphere publication in which they described progressive resistance of CTX-M-14 containing *K. pneumoniae* isolates. They were specifically interested in what they call P2 which was an extrachromosomal piece of DNA that contained an IS26 pseudo-compound transposon that was stably inherited despite lacking any clear replicon machinery. The circular P2 piece of genetic material (called cP2) was identical to ~35 kb piece present on a 220 kb plasmid called P1. IN figure 1, they provide a schema of P1 and P2 and show that in later, MDR isolates, P2 copy number, but not P1, is increased. They used ONT sequencing to examine mechanism of this amplification and deduce that it occurred due to tandem amplification of the ~35 kb piece present on P1 (read details are provided in Table 1). In figure 2, they use PFGE analysis of extra-chromosomal DNA focusing primarily on isolate 4-2 which is MDR and conclude that the cP2 could exist in a mono-, di-, or trimeric form. Figure 3 used qPCR and PFGE to show that amplification of P2 following cefotaxime exposure occurs on P1 and in cP2 and can persist after antimicrobial removal. Figure 4 shows that this amplification mechanism varied for different colonies. For Figure 5, they used transformation of genetic material into *recA*-deficient *E. coli* and showed that amplified pP2 can exist independently of *recA* but that cP2 required *recA*. Using wild-type and $\Delta recA$ strains, they show that changing the copy numbers of P2 only happens in the wild-type but not $\Delta recA$ background. Figure 6 is a summary figure that shows that the authors believe that amplification of P2 can start on P1 and then generate cP2 via *recA* which can then integrate with P1 again generating further amplification (or de-amplification).

Overall, the authors are addressing the important topic of mechanisms behind IS26 mediated amplification of AMR elements. The key finding is that AMR elements can be present both in a classical, plasmid context as well as in a circularized form lacking traditional plasmid elements. Both of these locations can both serve as places for amplification of AMR genes. To my knowledge, this is novel information and important for understanding bacterial adaptation to antimicrobials.

Their methodologies are sound, their conclusions are appropriate, and the manuscript is well written and clear (even though they are addressing a confusing topic). With their long-read data, researchers should be able to reproduce this work – indeed, we have looked at our raw, long-read data and seen similar closed elements. I thought their use of different restriction enzymes to distinguish between P1 and P2 was clever and that their use of *RecA* knockout helped define how these AMR elements are moving. My one criticism is that all of this is occurring in a single set of strains and perhaps the authors could screen other MDR organisms which they have done long read sequencing on to see whether they detect similar, circularized Tus.

Specific suggestions:

1. Table 1 is pretty confusing. Might be better to try to have this in some kind of simplified picture and then move the actual reads to the supplement.
2. In the paragraph starting on line 278, the figures move back and forth between figure 4 and figure 5. I think they should all be figure 4. Please check.

Reviewer #1 (Remarks to the Author):

This article describes the dynamics of genome amplification processes in bacteria, specifically focusing on the role of IS26 in the formation of pseudocompound transposons (PCTn) in multidrug-resistant regions of gram-negative bacteria. The study delves into the details of how these transposons can be tandemly amplified and presents evidence supporting the generation of a circular intermediate known as the "translocatable unit (TU)". The work is set within the context of *Klebsiella pneumoniae* multidrug-resistant clones.

The text, in general, is well-written, but a thorough proofreading is recommended to eliminate any minor grammatical issues.

The M&M section is well-structured and provides a comprehensive overview of the experimental procedures. Researchers familiar with the field should find it straightforward to reproduce the experiments based on this description. Indeed, the use of various techniques, including PCR, long-read sequencing, and pulsed-field gel electrophoresis (PFGE), enhances the credibility of the findings and shows a comprehensive approach to the study.

Moreover, the article describes a real-world scenario where *Klebsiella pneumoniae* isolates were obtained from a patient undergoing heavy antimicrobial treatment. This context enriches the findings by situating them within an applicable healthcare framework.

However, this manuscript has a lack of novelty in key findings. This work primarily focuses on the circular intermediate and tandem amplifications and both of these phenomena have already been described in previous studies, thus, the contribution of the present manuscript to the field might be limited. For instance, this manuscript did not provide direct evidence or demonstration of the transposition of the PCTn. Transposition is a crucial molecular mechanism that underlies the mobility of genetic elements, and its direct demonstration could have added significant value to the study. The absence of this evidence might be a potential weakness, especially if the title or the objectives of the study imply a focus on understanding the movement of the PCTn.

The authors presents a proposed model (Figure 6) that offers a coherent mechanism for the amplification and maintenance of P2, emphasizing the roles of antimicrobial pressure and RecA-dependent homologous recombination. However, there are gaps and uncertainties in the model, such as the exact mechanism of initial amplification and the potential role of IS26-associated TUs. These gaps can serve as directions for future research.

If the manuscript aims for a top-tier journal, addressing these gaps and uncertainties, even if only theoretically, might strengthen the paper. If experimental validation isn't possible in the current study, a comprehensive discussion of the implications of these gaps and their potential impacts on the model would be beneficial.

In conclusion, despite this weakness, this study provides a new contribution to the understanding of genome amplification in bacteria, especially in the context of antimicrobial resistance. The findings presented are well-supported by rigorous methods, and they open new avenues for further exploration in the mechanisms related with IS26 transposition that might be untangled in further studies.

<Our response>

Thank you for taking your valuable time to review our manuscript and for providing helpful comments that have assisted us in improving it, as described below.

As pointed out by the reviewer, the weakness of our study was that it is unknown whether and how IS26 TPase is involved in the amplification of P2, especially the initial amplification of pP2. We therefore constructed small single-copy plasmids carrying a “mini-PCTn” or its TPase-deficient derivative and introduced them into an *E. coli* strain and its RecA-deficient mutant (please see Figure 7 in the revised manuscript). By analyzing the copy numbers of the *bla*_{CTX-M-14} gene which corresponds to the PS in the original PCTn and performed FIGE-based profiling of the segments corresponding to the pP2 and cP2 (mini-pP2 and mini-cP2, respectively), we obtained not only the findings that support the notion that RecA-dependent HR plays a major role in the tandem amplification of mini-pP2 in the presence of antimicrobial selection pressure but also those indicating the presence of an unknown RecA/TPase-independent mechanism that generates the amplification of mini-pP2. So, the tandem amplification, at least the initial amplification, can occur in a RecA-dependent or independent manner. These results were described in a subsection newly added to the Results section (Lines 370-449) and the related data are shown in Figures 7, S4 and S5 and Table S1. Related parts in Abstract (Lines 38-39), Introduction (Lines 103-105), and Discussion (Lines 505-510, 513-514, and 523-529) have been modified according to these results. The methodological information related to this series of experiments has been described in the Materials and Methods (Lines 563-566, 568-582, 588, 590-592, 613-614, 624-640, and 673-677) and shown in Table S1.

It is still unknown whether and how IS26 TPase is involved in the tandem amplification of pP2. However, to address these issues, we need to identify the RecA/TPase-independent mechanism and then examine whether the tandem amplification of pP2 (or the mini-pP2) and its TPase-deficient derivative can occur under the genetic background where RecA is inactive and the putative RecA/TPase-independent mechanism does not work. We believe that these are important and interesting topics for our next project. We hope that the reviewer understands this situation.

Reviewer #2 (Remarks to the Author):

Herein the authors build upon a 2021 MSphere publication in which they described progressive resistance of CTX-M-14 containing *K. pneumoniae* isolates. They were specifically interested in what they call P2 which was an extrachromosomal piece of DNA that contained an IS26 pseudo-compound transposon that was stably inherited despite lacking any clear replicon machinery. The circular P2 piece of genetic material (called cP2) was identical to ~35 kb piece present on a 220 kb plasmid called P1. IN figure 1, they provide a schema of P1 and P2 and show that in later, MDR isolates, P2 copy number, but not P1, is increased. They used ONT sequencing to examine mechanism of this amplification and deduce that it occurred due to tandem amplification of the ~35 kb piece present on P1 (read details are provided in Table 1). In figure 2, they use PFGE analysis of extra-chromosomal DNA focusing primarily on isolate 4-2 which is MDR and conclude that the cP2 could exist in a mono-, di-, or trimeric form. Figure 3 used qPCR and PFGE to show that amplification of P2 following cefotaxime exposure occurs on P1 and in cP2 and can persist after antimicrobial removal. Figure 4 shows that this amplification mechanism varied for different colonies. For Figure 5, they used transformation of genetic material into *recA*-deficient *E. coli* and showed that amplified pP2 can exist independently of *recA* but that cP2 required *recA*. Using wild-type and $\Delta recA$ strains, they show that changing the copy numbers of P2 only happens in the wild-type but not $\Delta recA$ background. Figure 6 is a summary figure that shows that the authors believe that amplification of P2 can start on P1 and then generate cP2 via *recA* which can then integrate with P1 again generating further amplification (or de-amplification).

Overall, the authors are addressing the important topic of mechanisms behind IS26 mediated amplification of AMR elements. The key finding is that AMR elements can be present both in a classical, plasmid context as well as in a circularized form lacking traditional plasmid elements. Both of these locations can both serve as places for amplification of AMR genes. To my knowledge, this is novel information and important for understanding bacterial adaptation to antimicrobials.

Their methodologies are sound, their conclusions are appropriate, and the manuscript is well written and clear (even though they are addressing a confusing topic). With their long-read data, researchers should be able to reproduce this work – indeed, we have looked at our raw, long-read data and seen similar closed elements. I thought their use of different restriction enzymes to distinguish between P1 and P2 was clever and that their use of *RecA* knockout helped define how these AMR elements are moving. My one criticism is that all of this is occurring in a single set of strains and perhaps the authors could screen other MDR organisms which they have done long read sequencing on to see whether they detect similar, circularized Tus.

<Our response>

Thank you for spending your valuable time to review our manuscript and for your helpful comments. We agree that it is important to find similar, circularized TUs in other bacteria. We therefore screened the MDR bacteria that are now available in our laboratory but did not find such cases. So, we searched for literature that reported the results of similar long-read sequencing analyses. Through this search, we found one study (Hubbard et al., *Nat Commun.* 2020. doi: 10.1038/s41467-020-18668-2; Ref. 10 in the revised manuscript), in which similar long-read sequencing analysis was performed for two isogenic *E. coli* isolates (isolates 190693 and 169757): the former was susceptible to piperacillin/tazobactam and the latter one was resistant to it due to tandem amplification of a part of PTn6762, which the authors called the region tandemly amplified “TU”. We downloaded their long-read sequences (accession no; SRR12145915 and SRR12145917, respectively) and analyzed ≥ 2 kb reads that were mapped to their genome sequences, as we performed in our study. As summarized in the figure shown below, while tandemly repeated TU were not detected in the former isolate, long-reads derived from TUs that were tandemly repeated up to four times were detected in the latter isolate. Unfortunately, however, to determine whether these reads were derived from circularized multimeric TU or tandemly amplified TU in the chromosome, additional experiments, such as PFGE, need to be done. Therefore, we currently can not provide examples where circularized TUs can be detected. However, we believe that similar, circularized TUs will be identified by careful examination of IS26-associated PCTn in the future. We hope that the reviewer understands this situation on this matter.

Configuration pattern	Copy number of PS	SRR12145915	SRR12145917
		10 / 2,742	43 / 2,405
(1) Reads from chromosome			
	≥ 1	5	3
	2	0	1
	≥ 2	0	2
	≥ 1	5	3
	≥ 1	0	1
	≥ 2	0	2
(2) Reads from chromosome or TU			
	≥ 1	0	12
	≥ 1	0	1
	≥ 1	0	1
	≥ 1	0	11
	≥ 2	0	1
	≥ 2	0	2
	≥ 3	0	1
	≥ 3	0	1
	≥ 4	0	1

Specific suggestions:

1. Table 1 is pretty confusing. Might be better to try to have this in some kind of simplified picture and then move the actual reads to the supplement.

<Our response>

According to the reviewer's suggestion, we prepared a new figure (Figure 2 in the revised manuscript) that summarized the data shown in Table 1 of the original manuscript. We hope that with this figure, it is more easily understand what kinds of long read data (and how reads of each type) were obtained from each isolate.

2. In the paragraph starting on line 278, the figures move back and forth between figure 4 and figure

5. I think they should all be figure 4. Please check.

<Our response>

Thank you for pointing out these mistakes. We have corrected the figure numbers (Lines 291 and 303).

以下、査読コメント原文

Editor summary;

1. Please address reviewer 1's comment pertaining the the role of transposition and acknowledging the gaps in your current model.
2. Please follow reviewer 2's comments and clarify some of figures and table.
3. Also R2 - A look into long-read sequencing data other MDR bacteria for other circular Tus would strengthen the manuscript a lot.

Reviewer #1 (Remarks to the Author):

This article describes the dynamics of genome amplification processes in bacteria, specifically focusing on the role of IS26 in the formation of pseudocompound transposons (PCTn) in multidrug-resistant regions of gram-negative bacteria. The study delves into the details of how these transposons can be tandemly amplified and presents evidence supporting the generation of a circular intermediate known as the "translocatable unit (TU)". The work is set within the context of *Klebsiella pneumoniae* multidrug-resistant clones.

The text, in general, is well-written, but a thorough proofreading is recommended to eliminate any minor grammatical issues.

The M&M section is well-structured and provides a comprehensive overview of the experimental procedures. Researchers familiar with the field should find it straightforward to reproduce the experiments based on this description. Indeed, the use of various techniques, including PCR, long-read sequencing, and pulsed-field gel electrophoresis (PFGE), enhances the credibility of the findings and shows a comprehensive approach to the study.

Moreover, the article describes a real-world scenario where *Klebsiella pneumoniae* isolates were obtained from a patient undergoing heavy antimicrobial treatment. This context enriches the findings by situating them within an applicable healthcare framework.

However, this manuscript has a lack of novelty in key findings. This work primarily focuses on the circular intermediate and tandem amplifications and both of these phenomena have already been described in previous studies, thus, the contribution of the present manuscript to the field might be

limited. For instance, this manuscript did not provide direct evidence or demonstration of the transposition of the PCTn. Transposition is a crucial molecular mechanism that underlies the mobility of genetic elements, and its direct demonstration could have added significant value to the study. The absence of this evidence might be a potential weakness, especially if the title or the objectives of the study imply a focus on understanding the movement of the PCTn.

The authors presents a proposed model (Figure 6) that offers a coherent mechanism for the amplification and maintenance of P2, emphasizing the roles of antimicrobial pressure and RecA-dependent homologous recombination. However, there are gaps and uncertainties in the model, such as the exact mechanism of initial amplification and the potential role of IS26-associated TUs. These gaps can serve as directions for future research.

If the manuscript aims for a top-tier journal, addressing these gaps and uncertainties, even if only theoretically, might strengthen the paper. If experimental validation isn't possible in the current study, a comprehensive discussion of the implications of these gaps and their potential impacts on the model would be beneficial.

In conclusion, despite this weakness, this study provides a new contribution to the understanding of genome amplification in bacteria, especially in the context of antimicrobial resistance. The findings presented are well-supported by rigorous methods, and they open new avenues for further exploration in the mechanisms related with IS26 transposition that might be untangled in further studies.

Reviewer #2 (Remarks to the Author):

Herein the authors build upon a 2021 MSphere publication in which they described progressive resistance of CTX-M-14 containing *K. pneumoniae* isolates. They were specifically interested in what they call P2 which was an extrachromosomal piece of DNA that contained an IS26 pseudo-compound transposon that was stably inherited despite lacking any clear replicon machinery. The circular P2 piece of genetic material (called cP2) was identical to ~35 kb piece present on a 220 kb plasmid called P1. IN figure 1, they provide a schema of P1 and P2 and show that in later, MDR isolates, P2 copy number, but not P1, is increased. They used ONT sequencing to examine mechanism of this amplification and deduce that it occurred due to tandem amplification of the ~35 kb piece present on P1 (read details are provided in Table 1). In figure 2, they use PFGE analysis of extra-chromosomal DNA focusing primarily on isolate 4-2 which is MDR and conclude that the cP2 could exist in a mono-, di-, or trimeric form. Figure 3 used qPCR and PFGE to show that amplification of P2 following

cefotaxime exposure occurs on P1 and in cP2 and can persist after antimicrobial removal. Figure 4 shows that this amplification mechanism varied for different colonies. For Figure 5, they used transformation of genetic material into *recA*-deficient *E. coli* and showed that amplified pP2 can exist independently of *recA* but that cP2 required *recA*. Using wild-type and $\Delta recA$ strains, they show that changing the copy numbers of P2 only happens in the wild-type but not $\Delta recA$ background. Figure 6 is a summary figure that shows that the authors believe that amplification of P2 can start on P1 and then generate cP2 via *recA* which can then integrate with P1 again generating further amplification (or de-amplification).

Overall, the authors are addressing the important topic of mechanisms behind IS26 mediated amplification of AMR elements. The key finding is that AMR elements can be present both in a classical, plasmid context as well as in a circularized form lacking traditional plasmid elements. Both of these locations can both serve as places for amplification of AMR genes. To my knowledge, this is novel information and important for understanding bacterial adaptation to antimicrobials.

Their methodologies are sound, their conclusions are appropriate, and the manuscript is well written and clear (even though they are addressing a confusing topic). With their long-read data, researchers should be able to reproduce this work – indeed, we have looked at our raw, long-read data and seen similar closed elements. I thought their use of different restriction enzymes to distinguish between P1 and P2 was clever and that their use of *RecA* knockout helped define how these AMR elements are moving. **My one criticism is that all of this is occurring in a single set of strains and perhaps the authors could screen other MDR organisms which they have done long read sequencing on to see whether they detect similar, circularized Tus.**

Specific suggestions:

1. Table 1 is pretty confusing. Might be better to try to have this in some kind of simplified picture and then move the actual reads to the supplement.

2. In the paragraph starting on line 278, the figures move back and forth between figure 4 and figure

5. I think they should all be figure 4. Please check.

REVIEWERS' COMMENTS:

Reviewer #1 (Remarks to the Author):

Dear Authors,

I appreciate the comprehensive response and the significant efforts you have made to address the concerns I raised in my initial review. Your experiments involving small single-copy plasmids to study the amplification of "mini-PCTn" and its TPase-deficient derivative have substantially strengthened the manuscript. The new findings related to RecA-dependent and independent mechanisms for tandem amplification offer valuable insights and enrich the study's contribution to our understanding of IS26's role in genome amplification within the context of antimicrobial resistance.

Considering the improvements made to the manuscript and the substantial new data presented, I believe the manuscript now makes a significant contribution to our understanding of the dynamics of genome amplification processes in bacteria, particularly in the context of antimicrobial resistance. Therefore, I find the manuscript suitable for publication and look forward to seeing the impact of your work on the field.

Reviewer #2 (Remarks to the Author):

I believe the authors have responded appropriately to the reviewers' suggestions and concerns. I have no further suggestions.